# Are single-peaked tuning curves tuned for speed rather than accuracy?

**Movitz Lenninger[1]\*, Mikael Skoglund[1], Pawel Andrzej Herman[2], Arvind Kumar[2,3]\***

[1]Division of Information Science and Engineering, KTH Royal Institute of Technology, Stockholm, Sweden; [2]Division of Computational Science and Technology, KTH Royal Institute of Technology, Stockholm, Sweden; [3]Science for Life Laboratory, Stockholm, Sweden

**Abstract** According to the efficient coding hypothesis, sensory neurons are adapted to provide maximal information about the environment, given some biophysical constraints. In early visual areas, stimulus-induced modulations of neural activity (or tunings) are predominantly single-peaked. However, periodic tuning, as exhibited by grid cells, has been linked to a significant increase in decoding performance. Does this imply that the tuning curves in early visual areas are sub-optimal? We argue that the time scale at which neurons encode information is imperative to understand the advantages of single-peaked and periodic tuning curves, respectively. Here, we show that the possibility of catastrophic (large) errors creates a trade-off between decoding time and decoding ability. We investigate how decoding time and stimulus dimensionality affect the optimal shape of tuning curves for removing catastrophic errors. In particular, we focus on the spatial periods of the tuning curves for a class of circular tuning curves. We show an overall trend for minimal decoding time to increase with increasing Fisher information, implying a trade-off between accuracy and speed. This trade-off is reinforced whenever the stimulus dimensionality is high, or there is ongoing activity. Thus, given constraints on processing speed, we present normative arguments for the existence of the single-peaked tuning organization observed in early visual areas.

## Editor's evaluation

This fundamental study provides important insight into coding strategies in sensory areas. The study was well done, and the analysis and simulations were highly convincing. This study should be of particular interest to anybody who cares about efficient coding.

**\*For correspondence:**
movitzle@kth.se (ML);
arvkumar@kth.se (AK)

**Competing interest:** The authors declare that no competing interests exist.

## Introduction

One of the fundamental problems in systems neuroscience is understanding how sensory information can be represented in the spiking activity of an ensemble of neurons. The problem is exacerbated by the fact that individual neurons are highly noisy and variable in their responses, even to identical stimuli (*Arieli et al., 1996*). A common feature of early sensory representation is that the neocortical neurons in primary sensory areas change their average responses only to a small range of features of the sensory stimulus. For instance, some neurons in the primary visual cortex respond to moving bars oriented at specific angles (*Hubel and Wiesel, 1962*). This observation has led to the notion of *tuning curves*. Together, a collection of tuning curves provides a possible basis for a neural code.

A considerable emphasis has been put on understanding how the structure of noise and correlations affect stimulus representation given a set of tuning curves (*Shamir and Sompolinsky, 2004*; *Averbeck and Lee, 2006*; *Franke et al., 2016*; *Zylberberg et al., 2016*; *Moreno-Bote et al., 2014*; *Kohn et al., 2016*). More recently, the issue of local and catastrophic errors, dating back to the

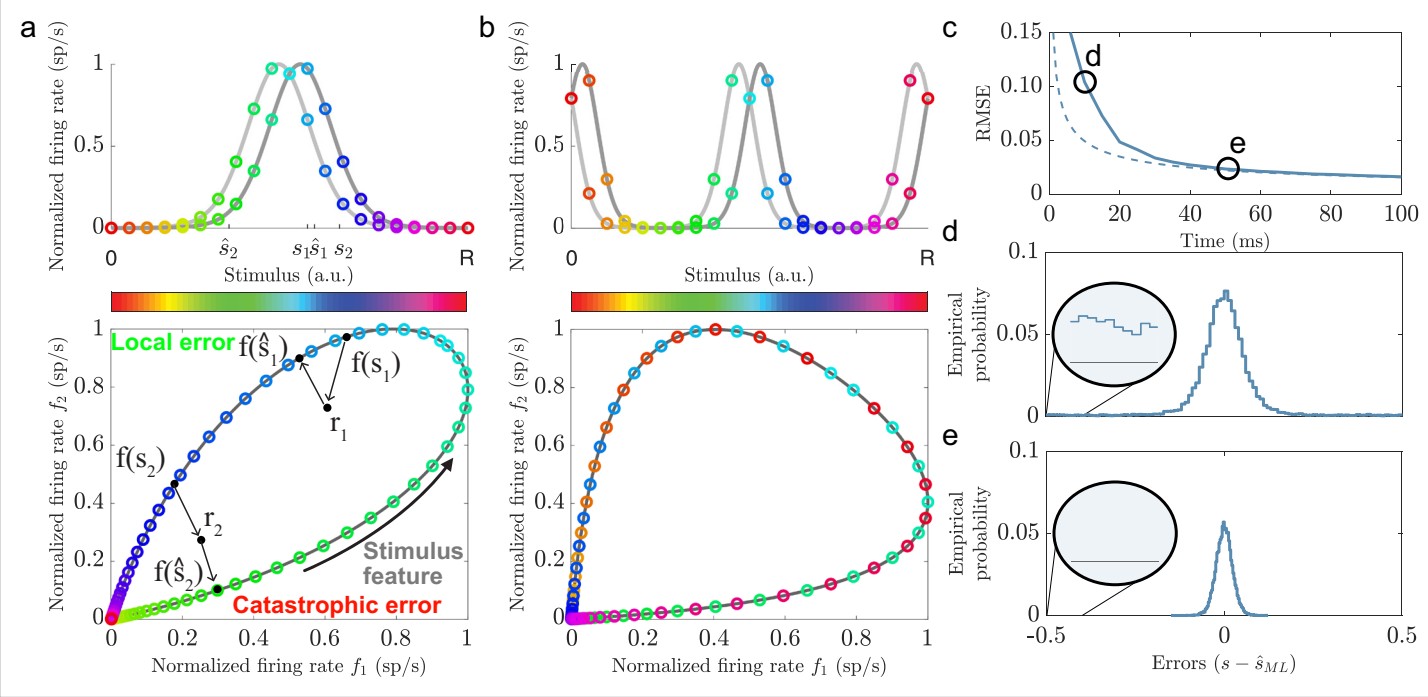

**Figure 1.** Illustrations of local and catastrophic errors. (**a**) Top: A two-neuron system encoding a single variable using single-peaked tuning curves ($\lambda = 1$). Bottom: The tuning curves create a one-dimensional activity trajectory embedded in a two-dimensional neural activity space (black trajectory). Decoding the two stimulus conditions, $s_1$ and $s_2$, illustrates the two types of estimation errors that can occur due to trial-by-trial variability, local ($\hat{s}_1$) and catastrophic ($\hat{s}_2$). (**b**) Same as in (**a**) but for periodic tuning curves ($\lambda = 0.5$). Notice that the stimulus conditions are intermingled and that the stimulus can not be determined from the firing rates. (**c**) Time evolution of the root mean squared error (RMSE) using maximum likelihood estimation (solid line) and the Cramér-Rao bound (dashed line) for a population of single-peaked tuning curves ($N = 600$, $w = 0.3$, average evoked firing rate $\overline{f_{stim}} = 20 \exp(-1/w)B_0(1/w)$ sp/s, and $b = 2$ sp/s). For about 50 ms the RMSE is significantly larger than the predicted lower bound. (**d**) The empirical error distributions for the time point indicated in (**c**), where the RMSE strongly deviates from the predicted lower bound. Inset: A non-zero empirical error probability spans the entire stimulus domain. (**e**) Same as in (**d**) when the RMSE roughly converges to the Cramér-Rao bound. Notice the absence of large estimation errors.

work of Shannon (**Shannon, 1949**), has been raised in the context of neuroscience (e.g. **Xie, 2002**; **Sreenivasan and Fiete, 2011**). Intuitively, local errors are small estimation errors that depend on the trial-by-trial variability of the neural responses and the local shapes of the tuning curves surrounding the true stimulus condition (**Figure 1a** bottom plot, see $s_1$). On the other hand, catastrophic errors are very large estimation errors that depend on the trial-by-trial variability and the global shape of the tuning curves (**Figure 1a** bottom plot, see $s_2$). While a significant effort has been put into studying how stimulus tuning and different noise structures affect local errors, less is known about the interactions with catastrophic errors. For example, *Fisher information* is a common measure of the accuracy of a neural code (**Brunel and Nadal, 1998**; **Abbott and Dayan, 1999**; **Guigon, 2003**; **Moreno-Bote et al., 2014**; **Benichoux et al., 2017**). The Cramér-Rao bound states that a lower limit of the minimal mean squared error (MSE) for any unbiased estimator is given by the inverse of Fisher information (**Lehmann and Casella, 1998**). Thus, increasing Fisher information reduces the lower bound on MSE. However, because Fisher information can only capture local errors, the true MSE might be considerably larger in the presence of catastrophic errors (**Xie, 2002**; **Kostal et al., 2015**; **Malerba et al., 2022**), especially if the available decoding time is short (**Bethge et al., 2002**; **Finkelstein et al., 2018**).

A curious observation is that the tuning curves in early visual areas predominately use single-peaked firing fields, whereas grid cells in the entorhinal cortex are known for their periodically distributed firing fields (**Hafting et al., 2005**). It has been shown that the multiple firing locations of grid cells increase the precision of the neural code compared to single-peaked tuning curves (**Sreenivasan and Fiete, 2011**; **Mathis et al., 2012**; **Wei et al., 2015**). This raises the question of why periodic firing fields are not a prominent organization of early visual processing too?

The theoretical arguments in favor of periodic tuning curves have mostly focused on local errors under the assumption that catastrophic errors are negligible (*Sreenivasan and Fiete, 2011*). However, given the response variability, it takes a finite amount of time to accumulate a sufficient number of spikes to decode the stimulus. Given that fast processing speed is a common feature of visual processing (*Thorpe et al., 1996*; *Fabre-Thorpe et al., 2001*; *Rolls and Tovee, 1994*; *Resulaj et al., 2018*), it is crucial that each neural population in the processing chain can quickly produce a reliable stimulus-evoked signal. Therefore, the time required to produce signals without catastrophic errors will likely put fundamental constraints on any neural code, especially in early visual areas.

Here, we contrast Fisher information with the minimal decoding time required to remove catastrophic errors (i.e. the time until Fisher information becomes a reasonable descriptor of the MSE). We base the results on the maximum likelihood estimator for uniformly distributed stimuli (i.e., the maximum a posteriori estimator) using populations of tuning curves with different numbers of peaks. We show that the minimal decoding time tends to increase with increasing Fisher information in the case of independent Poissonian noise to each neuron. This suggests a trade-off between the decoding accuracy of a neural population and the speed by which it can produce a reliable signal. Furthermore, we show that the difference in minimal decoding time grows with the number of jointly encoded stimulus features (stimulus dimensionality) and in the presence of ongoing (non-specific) activity. Thus, single-peaked tuning curves require shorter decoding times and are more robust to ongoing activity than periodic tuning curves. Finally, we illustrate the issue of large estimation errors and periodic tuning in simple spiking neural network model tracking either a step-like stimulus change or a continuously time-varying stimulus.

## Results
### Shapes of tuning curves, Fisher information, and catastrophic errors

To enable a comparison between single-peaked and periodic (multi-peaked) tuning curves, we consider circular tuning curves responding to a D-dimensional stimulus, **s**, according to

$$f_i(\mathbf{s}) = a_i \prod_{j=1}^{D} \exp\left(\frac{1}{w}\left(\cos\left(\frac{2\pi}{\lambda_i}(s_j - s'_{i,j})\right) - 1\right)\right) + b \tag{1}$$

where $a_i$ is the peak amplitude of the stimulus-related tuning curve $i$, $w$ is a width scaling parameter, $\lambda_i$ defines the spatial period of the tuning curve, $s'_{i,j}$ determines the location of the peak(s) in the $j$:th stimulus dimension, and $b$ determines the amount of ongoing activity (see *Figure 1a–b*, top panels). The parameters are kept fixed for each neuron, thus ignoring any effect of learning or plasticity. In the following, the stimulus domain is set to $\mathbf{s} \in [0, 1)^D$ for simplicity. To avoid boundary effects, we assume that the stimulus has periodic boundaries (i.e. $s_j = 0$ and $s_j = 1$ are the same stimulus condition) and adjust any decoded value to lie within the stimulus domain, for example,

$$\hat{s}_{ML} = 1 + 0.1 \ (\text{mod } 1) = 0.1, \tag{2}$$

see Materials and methods - 'Implementation of maximum likelihood estimator' for details.

We assume that the stimulus is uniformly distributed across its domain and that its dimensions are independent. This can be seen as a worst-case scenario as it maximizes the entropy of the stimulus. In a single trial, we assume that the number of emitted spikes for each neuron is conditionally independent, and follows a Poisson distribution, given some stimulus-dependent rate $f_i(s)$. Thus, the probability of observing a particular activity pattern, **r**, in a population of $N$ neurons given the stimulus-dependent rates and decoding time, $T$, is

$$p(\mathbf{r}|\mathbf{s}, T) = \prod_{i=1}^{N} p(r_i|Tf_i(\mathbf{s})) = \prod_{i=1}^{N} \frac{(Tf_i(\mathbf{s}))^{r_i} \exp(-Tf_i(\mathbf{s}))}{r_i!}. \tag{3}$$

Given a model of neural responses, the Cramér-Rao bound provides a lower bound on the accuracy by which the population can communicate a signal as the inverse of the Fisher information. For sufficiently large populations, using the population and spike count models in *Equation 1* and *Equation 3*, Fisher information is given by (for $a_i = a$ and $b = 0$ for all neurons, see *Sreenivasan and Fiete, 2011* or Appendix 2 - 'Fisher information and the Cramér-Rao bound' for details)

$$J \approx (2\pi)^2 \frac{aTN}{w} B_0(1/w)^{D-1} B_1(1/w) \exp(-D/w)\overline{\lambda^{-2}} \tag{4}$$

where $\overline{\lambda^{-2}}$ denotes the sample average of the squared inverse of the (relative) spatial periods across the population, and $B_\alpha(\cdot)$ denotes the modified Bessel functions of the first kind. *Equation 4* (and similar expressions) suggests that populations consisting of periodic tuning curves, for which $\overline{\lambda^{-2}} \gg 1$, are superior at communicating a stimulus signal than a population using tuning curves with only single peaks, where $\overline{\lambda^{-2}} = 1$. However, (inverse) Fisher information only predicts the amount of local errors for an efficient estimator. Hence, the presence of catastrophic errors (*Figure 1a*, bottom) can be identified by large deviations from the predicted MSE for an asymptotically efficient estimator (*Figure 1c–d*). Therefore, we define minimal decoding time as the shortest time required to approach the Cramér-Rao bound (*Figure 1c and e*).

## Periodic tuning curves and stimulus ambiguity

To understand why the amount of catastrophic error can differ with different spatial periods, consider first the problem of stimulus ambiguity that can arise with periodic tuning curves. If all tuning curves in the population share the same relative spatial period, $\lambda$, then the stimulus-evoked responses can only provide unambiguous information about the stimulus in the range $[0, \lambda)$. Beyond this range, the response distributions are no longer unique. Thus, single-peaked tuning curves ($\lambda = 1$) provide unambiguous information about the stimulus. Periodic tuning curves ($\lambda < 1$), on the other hand, require the use of tuning curves with two or more distinct spatial periods to resolve the stimulus ambiguity (*Fiete et al., 2008*; *Mathis et al., 2012*; *Wei et al., 2015*). In the following, we assume the tuning curves are organized into discrete modules, where all tuning curves within a module share spatial period (*Figure 1b*) mimicking the organization of grid cells (*Stensola et al., 2012*). For convenience, assume that $\lambda_1 > \lambda_2 > ... > \lambda_L$ where $L$ is the number of modules. Thus, the first module provides the most coarse-grained resolution of the stimulus interval, and each successive module provides an increasingly fine-grained resolution. It has been suggested that a geometric progression of spatial periods, such that $\lambda_i = c\lambda_{i-1}$ for some spatial factor $0 < c \leq 1$, may be optimal for maximizing the resolution of the stimulus while reducing the required number of neurons (*Mathis et al., 2012*; *Wei et al., 2015*). However, trial-by-trial variability can still cause stimulus ambiguity and catastrophic errors - at least

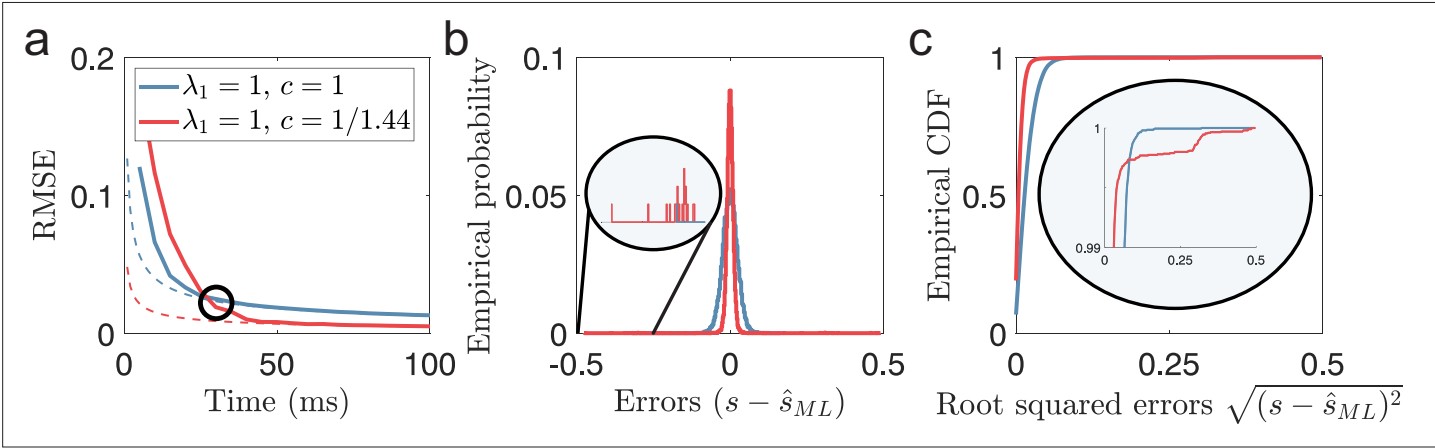

**Figure 2.** (Very) Short decoding times when both Fisher information and MSE fails. (**a**) Time evolution of root mean squared error (RMSE), averaged across trials and stimulus dimensions, using maximum likelihood estimation (solid lines) for two populations (blue: $\lambda_1 = 1$, $c = 1$, red: $\lambda_1 = 1$, $c = 1/1.44$). Dashed lines indicate the lower bound predicted by Cramér-Rao. The black circle indicates the point where the periodic population has become optimal in terms of MSE. (**b**) The empirical distribution of errors for the time indicated by the black circle in (**a**). The single-peaked population (blue) has a wider distribution of errors centered around 0 compared to the periodic population (red), as suggested by having a higher MSE. Inset: Zooming in on rare error events reveals that while the periodic population has a narrower distribution of errors around 0, it also has occasional errors across large parts of the stimulus domain. (**c**) The empirical CDF of the errors for the same two populations as in (**b**). Inset: a zoomed-in version (last 1%) of the empirical CDF highlights the heavy-tailed distribution of errors for the periodic population. Parameters used in the simulations: stimulus dimensionality $D = 2$, the number of modules $L = 5$, number of neurons $N = 600$, average evoked firing rate $\overline{f_{stim}} = 20 \exp(-1/w)B_0(1/w)$ sp/s, ongoing activity $b = 2$ sp/s, and width parameter $w = 0.3$. Note that the estimation errors for the two stimulus dimensions are pooled together.

for short decoding times, as we show later, even when using multiple modules with different spatial periods.

## (Very) Short decoding times - when both Fisher information and MSE fails

While it is known that Fisher information is not an accurate predictor of the MSE when the decoding time is short (**Bethge et al., 2002**), less has been discussed about the issue of MSE. Although MSE is often interpreted as a measure of accuracy, its insensitivity to rare outliers makes it a poor measure of reliability. Therefore, comparing MSE directly between populations can be a misleading measure of reliability if the distributions of errors are qualitatively different. If the amounts of local errors differ, lower MSE does not necessarily imply fewer catastrophic errors. This is exemplified in **Figure 2**, comparing a single-peaked and a periodic population encoding a two-dimensional stimulus using the suggested optimal scale factor, $c \approx 1/1.44$ (**Wei et al., 2015**). During the first $\approx 30$ ms, the single-peaked population has the lowest MSE of the two populations despite having lower Fisher information (**Figure 2a**). Furthermore, comparing the error distribution after the periodic population achieves a lower MSE (the black circle in **Figure 2a**) shows that the periodic population still suffers from rare errors that span the entire stimulus range (**Figure 2b–c**, insets). As we will show, a comparison of MSE, as a measure of reliability, only becomes valid once catastrophic errors are removed. Here we assume that catastrophic errors should strongly affect the usability of a neural code. Therefore, we argue that the first criterion for any rate-based neural code should be to satisfy its constraint on decoding time to avoid catastrophic errors.

## Minimal decoding times in populations with two modules

How does the choice of spatial periods impact the required decoding time to remove catastrophic errors? To get some intuition, we first consider the case of populations encoding a one-dimensional stimulus using only two different spatial scales, $\lambda_1$ and $\lambda_2$. From the perspective of a probabilistic decoder (**Seung and Sompolinsky, 1993**; **Deneve et al., 1999**; **Ma et al., 2006**), assuming that the stimulus is uniformly distributed, the maximum likelihood (ML) estimator is Bayesian optimal (and asymptotically efficient). The maximum likelihood estimator aims at finding the stimulus condition which is the most likely cause of the observed activity, $\mathbf{r}$, or

$$\hat{s}_{ML} = \arg\max_s p(\mathbf{r}|s), \tag{5}$$

where $p(\mathbf{r}|s)$ is called the likelihood function. The likelihood function equals the probability of observing the observed neural activity, $\mathbf{r}$, assuming that the stimulus condition was $s$. In the case of independent Poisson spike counts (or at least independence across modules), each module contributes to the joint likelihood function $p(\mathbf{r}|s)$ with individual likelihood functions, $Q_1$ and $Q_2$ (**Wei et al., 2015**). Thus, the joint likelihood function can be seen as the product of the two individual likelihood functions, where each likelihood is $\lambda_i$-periodic

$$p(\mathbf{r}|s) = Q_1(\mathbf{r}|s)Q_2(\mathbf{r}|s). \tag{6}$$

In this sense, each module provides its own ML-estimate of the stimulus, $s_{ML}^{(1)} = \arg\max_s Q_1(\mathbf{r}|s)$ and $s_{ML}^{(2)} = \arg\max_s Q_2(\mathbf{r}|s)$. Because of the periodicity of the tuning curves, there can be multiple modes for each of the likelihoods (e.g. **Figure 3a and b**, top panels). For the largest mode of the joint likelihood function to also be centered close to the true stimulus condition, the distance $\delta$ between $s_{ML}^{(1)}$ and $s_{ML}^{(2)}$ must be smaller than between any other pair of modes of $Q_1$ and $Q_2$. Thus, to avoid catastrophic errors, $\delta$ must be smaller than some largest allowed distance $\delta^*$ which guarantees this relation (see **Equations 25–30** for calculation of $\delta^*$ assuming the stimulus is in the middle of the domain). As $\delta$ varies from trial to trial, we limit the probability of the decoder experiencing catastrophic errors to some small error probability, $p_{error}$, by imposing that

$$\Pr(|\delta| > \delta^*) < p_{error}. \tag{7}$$

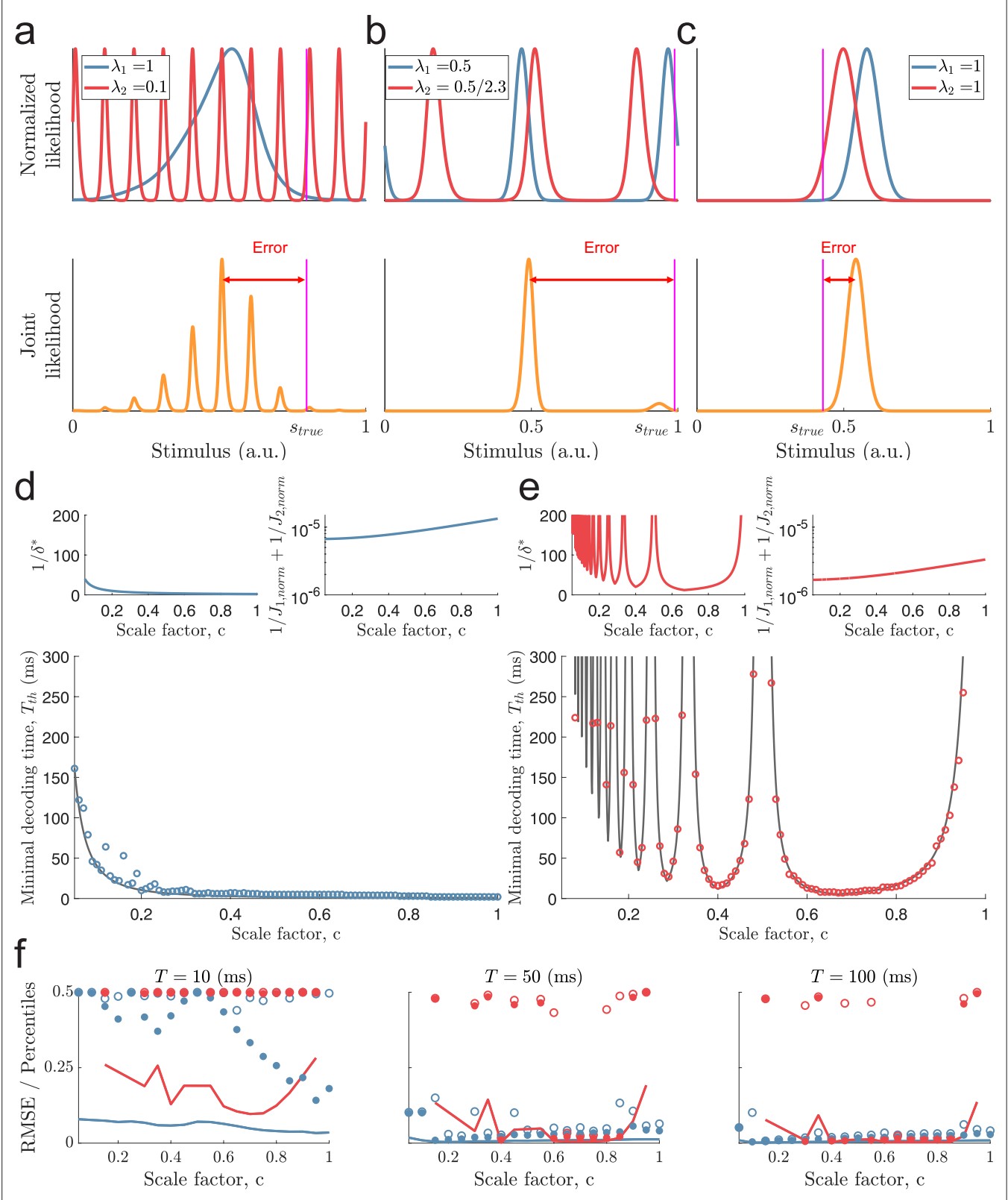

**Figure 3.** Catastophic errors and minimal decoding times in populations with two modules. (**a**) Top: Sampled individual likelihood functions of two modules with very different spatial periods. Bottom: The sampled joint likelihood function for the individual likelihood functions in the top panel. (**b**–**c**) Same as in (**a**) but for spatial periods that are similar but not identical and for a single-peaked population, respectively. (**d**) Bottom: The dependence of the scale factor c on the minimal decoding time for $\lambda_1 = 1$. Blue circles indicate the simulated minimal decoding times, and the black line indicates

*Figure 3 continued on next page*

*Figure 3 continued*

the estimation of the minimal decoding times according to **Equation 8**, with $p_{error} = 10^{-4}$. Top left: The predicted value of $1/\delta^*$. Top right: The inverse of the Fisher information. (**e**) Same as (**d**) but for $\lambda_1 = 1/2$. (**f**) RMSE (lines), the 99.8th percentile (filled circles), and the maximal error (open circles) of the error distribution for several choices of scale factor, $c$, and decoding time. The color code is the same as in panels (**d-e**). The parameters used in (**d-f**) are: population size $N = 600$, number of modules $L = 2$, scale factors $c = 0.05 - 1$, width parameter $w = 0.3$, average evoked firing rate $\overline{f_{stim}} = 20 \exp(-1/w)B_0(1/w)$ sp/s, ongoing activity $b = 0$ sp/s, and threshold factor $\alpha = 2$.

The online version of this article includes the following figure supplement(s) for figure 3:

**Figure supplement 1.** Example of decoding a stimulus close to the periodic edge.

**Figure supplement 2.** Same as (*Figure 2d–e*) but using threshold factor $\alpha = 1.2$.

Assuming that the estimation of each module becomes efficient before the joint estimation, **Equation 7** can be reinterpreted as a lower bound on the required decoding time before the estimation based on the joint likelihood function becomes efficient

$$T_{th} > 2 \left( \frac{\mathrm{erfinv}(1 - p_{error})}{\delta^*} \right)^2 \left( \frac{1}{J_{1,norm}} + \frac{1}{J_{2,norm}} \right), \tag{8}$$

where $\mathrm{erfinv}(\cdot)$ is the inverse of the error function and $J_{k,norm}$ refers to the time-normalized Fisher information of module $k$ (see Materials and methods for derivation). Thus, the spatial periods of the modules influence the minimal decoding time by determining: (1) the largest allowed distance $\delta^*$ between the estimates of the modules and (2) the variances of the estimations given by the inverse of their respective Fisher information.

To give some intuition of the approximation, if the spatial periods of the modules are very different, $\lambda_2 \ll \lambda_1$, then there exist many peaks of $Q_2$ around the peak of $Q_1$ (**Figure 3a**). Additionally, there can be modes of $Q_1$ and $Q_2$ far away from the true stimulus close together. Thus, $\lambda_2 \ll \lambda_1$ can create a highly multi-modal joint likelihood function where small deviations in $s_{ML}^{(1)}$ and $s_{ML}^{(2)}$ can cause a shift, or a change, of the maximal mode of the joint likelihood. To avoid this, $\delta^*$ must be small, leading to longer decoding times by **Equation 8**. Furthermore, suppose the two modules have similar spatial periods $\lambda_2 \sim \lambda_1$, or $\lambda_1$ is close to a multiple of $\lambda_2$. In that case, the distance between the peaks a few periods away is also small, again leading to longer decoding times (**Figure 3b**). In other words, periodic tuning suffers from the dilemma that small shifts in the individual stimulus estimates can cause catastrophic shifts in the joint likelihood function. Although these might be rare events, the possibility of such errors increases the probability of catastrophic errors. Thus, assuming $\lambda_1 < 1$, both small and large scale factors $c$ can lead to long decoding times. When $\lambda_1 = 1$, however, only small-scale factors $c$ pose such problems, at least unless the stimulus is close to the periodic edge (i.e. $s \approx 0$ or $s \approx 1$, see **Figure 3—figure supplement 1**). On the other hand, compared to single-peaked tuning curves, periodic tuning generally leads to sharper likelihood functions, increasing the accuracy of the estimates once catastrophic errors are removed (e.g., compare the widths of the joint likelihood functions in **Figure 3a–c**).

To test the approximation in **Equation 8**, we simulated a set of populations ($N = 600$ neurons) with different spatial periods. The populations were created using identical tuning parameters except for the spatial periods, whose distribution varied across the populations, and the amplitudes, which were adjusted to ensure an equal average firing rate (across all stimulus conditions) for all neurons (see Materials and methods for details on simulations). As described above, the spatial periods were related by a scale factor $c$. Different values of $c$ were tested for the largest period being either $\lambda_1 = 1$ or $\lambda_1 = 1/2$. Furthermore, only populations with unambiguous codes over the stimulus interval were included (i.e. $c \neq 1/2, 1/3, 1/4, \ldots$ for $\lambda_1 = 1/2$; **Mathis et al., 2012**). Note, however, that there is no restriction on the periodicity of the tuning curves to align with the periodicity of the stimulus (i.e. $1/\lambda_i$ does not need to be an integer). For each population, the minimal decoding time was found by gradually increasing the decoding time until the empirical MSE was lower than twice the predicted lower bound (i.e. $\alpha = 2$, see **Equation 10** and Materials and methods for details). Limiting the probability of catastrophic errors to $p_{error} = 10^{-4}$, **Equation 8** is a good predictor of the minimal decoding time (**Figure 3d–e**, bottom panels, coefficient of determination $R^2 \approx 0.92$ and $R^2 \approx 0.95$ for $\lambda_1 = 1$ and $\lambda_1 = 1/2$, respectively). For both $\lambda_1 = 1$ and $\lambda_1 = 1/2$, the minimal decoding time increases overall with decreasing scale factor, $c$ (see **Figure 3d–e**). However, especially for $\lambda_1 = 1/2$, the trend is interrupted

by large peaks (*Figure 3e*). For $\lambda_1 = 1$, there are deviations from the predicted minimal decoding time for small scale factors, $c$. They occur whenever $\lambda_2$ is slightly below a multiple of $\lambda_1 = 1$, and get more pronounced when increasing the sensitivity to the threshold factor $\alpha = 1.2$ (see *Figure 3—figure supplement 2*). We believe one cause of these deviations is the additional shifts across the periodic boundary (as in *Figure 3—figure supplement 1*) that can occur when $c$ is just below 1/2, 1/3, 1/4, ..., etc.

To confirm that the estimated minimal decoding times have some predictive power on the error distributions, we re-simulated a subset of the populations for various decoding times, $T$, using 15,000 randomly sampled stimulus conditions (*Figure 3f*). Both the RMSE and outlier errors (99.8th percentile and the maximal error, that is, 100th percentile) agree with the shape of minimal decoding times, suggesting that a single-peaked population is good at removing large errors at very short time scales.

## Minimal decoding times for populations with more than two modules

From the two-module case above, it is clear that the choice of scale factor influences the minimal decoding time. However, *Equation 8* is difficult to interpret and is only valid for two-module systems ($L = 2$). To approximate how the minimal decoding time scales with the distribution of spatial periods in populations with more than two modules, we extended the approximation method first introduced by Xie (*Xie, 2002*). The method was originally used to assess the number of neurons required to reach the Cramér-Rao bound for single-peaked tuning curves with additive Gaussian noise for the ML estimator. In addition, it only considered encoding a one-dimensional stimulus variable. We adapted this method to approximate the required decoding time for stimuli with arbitrary dimensions, Poisson-distributed spike counts, and tuning curves with arbitrary spatial periods. In this setting, the scaling of minimum decoding time with the spatial periods, $\lambda_1, ..., \lambda_L$, can be approximated as (see Materials and methods for derivation)

$$T_{th} \gg A(w)\frac{1}{aN}\frac{\exp(D/w)}{B_0(1/w)^{(D-1)}}\frac{\overline{\lambda^{-3}}^2}{\overline{\lambda^{-2}}^3} \simeq \frac{A^*(w)}{N\overline{f_{stim}(D)}}\frac{\overline{\lambda^{-3}}^2}{\overline{\lambda^{-2}}^3}, \tag{9}$$

where $\overline{\lambda^{-2}}$ and $\overline{\lambda^{-3}}$ indicate the sample average across the inverse spatial periods (squared or cubed, respectively) in the population, $\overline{f_{stim}(D)}$ is the average evoked firing rate across the stimulus domain, and $A(w)$ (or $A^*(w)$) is a function of $w$ (see Materials and methods for detailed expression). The last approximation holds with equality whenever all tuning curves have an integer number of peaks. The derivation was carried out assuming the absence of ongoing activity and that the amplitudes within each population are similar, $a_1 \approx ... \approx a_N$. Importantly, the approximation also assumes the existence of a unique solution to the maximum likelihood equations. Therefore, it is ill-equipped to predict the issues of stimulus ambiguity. Thus, going back to the two-module cases, *Equation 9* cannot capture the additional effects of $\lambda_2 \ll \lambda_1$ or when $\lambda_1$ is close to a multiple of $\lambda_2$, as in *Figure 3d–e*. On the other hand, complementing the theory presented in *Equation 8*, *Equation 9* provides a more interpretable expression of the scaling of minimal decoding time. For $c \leq 1$, the minimal decoding time, $T_{th}$, is expected to increase with decreasing scale factor, $c$ (see *Equation 47*). The scaling should also be similar for different choices of $\lambda_1$. Furthermore, assuming all other parameters are constant, the minimal decoding time should grow roughly exponentially with the number of stimulus dimensions.

To confirm the validity of *Equation 9*, we simulated populations of $N = 600$ tuning curves across $L = 5$ modules. Again, the spatial periods across the modules were related by a scale factor, $c$ (*Figure 4a*). To avoid the effects of $c \ll 1$, we limited the range of the scale factor to $0.3 \leq c \leq 1$. The upper bound on $c$ was kept (for $\lambda_1 = 1$) to include entirely single-peaked populations. Again, the assumption of homogeneous amplitudes in *Equation 9* was dropped in simulations (*Figure 4b*, left column) to ensure that the average firing rate across the stimulus domain is equal for all neurons (see *Figure 4b*, right column, for the empirical average firing rates). This had little effect on Fisher information, where the theoretical prediction was based on the average amplitudes across all populations with the same $\lambda_1$ and stimulus dimensionality $D$ (see *Figure 4c*, inset). As before, Fisher information grows with decreasing scale factor, $c$, and with decreasing spatial period $\lambda_1$. As expected, increasing the stimulus dimensionality decreases Fisher information if all other parameters are kept constant. On the other hand, the minimal decoding time increases with decreasing spatial periods and increases with stimulus dimensionality (*Figure 4c*). The increase in decoding time between $D = 1$ and $D = 2$ is also very well predicted by *Equation 9*, at least for $c > 0.5$ (*Figure 4—figure supplement 1a*). In these simulations, the choice of width parameter is compatible with experimental data (*Ringach et al.,*

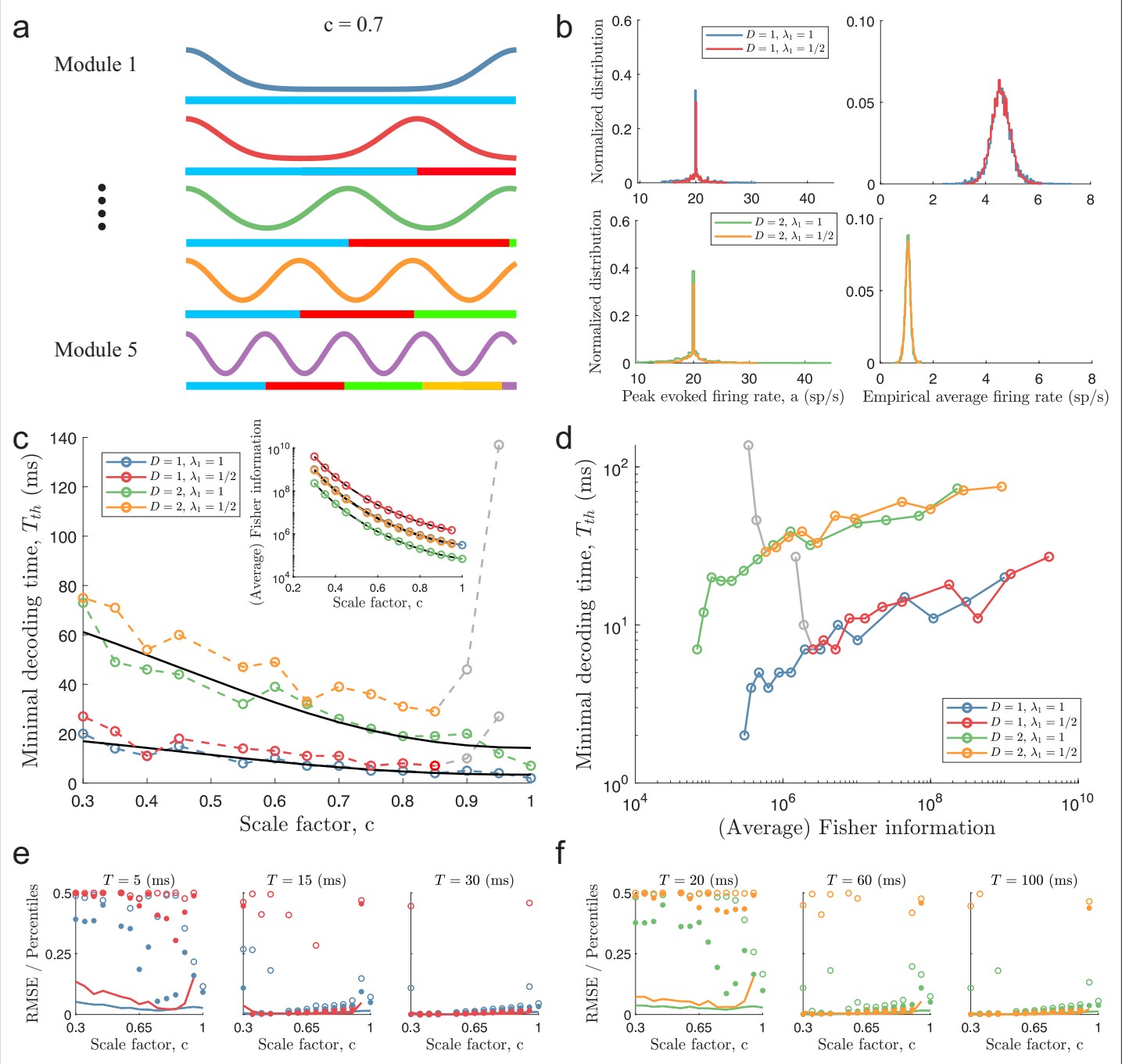

**Figure 4.** Minimal decoding times for populations with five modules. (**a**) Illustration of the likelihood functions of a population with $L = 5$ modules using scale factor $c = 0.7$. (**b**) The peak stimulus-evoked amplitudes of each neuron (left column) were selected such that all neurons shared the same expected firing rate for a given stimulus condition (right column). (**c**) Inset: Plot of average Fisher information as a function of the scale factor $c$ (colored lines: estimations from simulation data, black lines: theoretical approximations). Main plot: Plot of minimal decoding time as a function of scale factor $c$. Minimal decoding time tends to increase with decreasing grid scales (colored lines: estimated minimal decoding time from simulations, black lines: fitted theoretical predictions using *Equation 47*). The gray color corresponds to points with large discrepancies between the predicted and the simulated minimal decoding times. (**d**) Plot of the average Fisher information against the minimal decoding time. Points colored in gray are the same as in panel (**c**). (**e**) RMSE (lines), the 99.8$^{th}$ percentile (filled circles), and the maximal error (open circles) of the error distribution when decoding a 1-dimensional stimulus for several choices of decoding time. The color code is the same as above. (**f**) same as (**e**) but for a two-dimensional stimulus. Note that the error distributions across stimulus dimensions are pooled together. Parameters used in panels (**a-d**): population size $N = 600$, number of modules $L = 5$, scale factors $c = 0.3 - 1$, width parameter $w = 0.3$, average evoked firing rate $\overline{f_{stim}} = 20 \exp(-D/w)B_0(1/w)^D$ sp/s, ongoing activity $b = 0$ sp/s, and threshold factor $\alpha = 2$.

*Figure 4 continued on next page*

*Figure 4 continued*

The online version of this article includes the following figure supplement(s) for figure 4:

**Figure supplement 1.** Scaling of minimal decoding time with stimulus dimensionality and tuning width.

**Figure supplement 2.** Minimal decoding times using a different threshold factor.

**Figure supplement 3.** Minimal decoding times based on one-sided KS-tests.

**Figure supplement 4.** Time evolution of MSE and outlier errors.

**Figure supplement 5.** Spike counts required for removing catastrophic errors.

**Figure supplement 6.** Minimal decoding times for populations without scale factors.

*2002*), but similar trends were found for a range of different width parameters (although the differences become smaller for small $w$, see *Figure 4—figure supplement 1b–d*).

From *Equation 9*, we fitted two constants, $K_1$ (regressor) and $K_2$ (intercept), using least square regression across populations sharing the same largest period, $\lambda_1$, and stimulus dimensionality, $D$ (see *Equation 47*). Within the simulated range of scale factors, the regressions provide reasonable fits for the populations with $\lambda_1 = 1$ (*Figure 4c*, coefficient of determination $R^2 \approx 0.89$ and $R^2 \approx 0.90$ for $D = 1$ and $D = 2$, respectively). For the populations with $\lambda_1 = 1/2$, *Equation 9* becomes increasingly unable to predict the behavior of the minimal decoding time as $c$ approaches 1 (see the red and yellow lines in *Figure 4c–d*). On the other hand, as was suggested above, the scaling of the minimal decoding time with $c$ is, in fact, similar for $\lambda_1 = 1$ and $\lambda_1 = 1/2$ whenever $c$ is less than $\approx 0.9$. As suggested by *Figure 4d*, there is also a strong correlation between Fisher information and minimal decoding time, again indicating a speed-accuracy trade-off. Furthermore, similar results are obtained when either decreasing the threshold factor to $\alpha = 1.2$ (*Figure 4—figure supplement 2*) or changing the minimal decoding time criterion to a one-sided Kolmogorov–Smirnov test (KS-test) between the empirical distribution of errors and the Gaussian error distribution predicted by the Cramér-Rao bound (*Figure 4—figure supplement 3*, using an ad-hoc Bonferroni-type correction for multiple sequential testing, $\alpha/j$, where $j$ is the $j$ th time comparison and $\alpha = 0.05$ is the significance level.)

To further illustrate the relationship between minimum decoding time and the distribution of catastrophic errors, we re-simulated the same populations using fixed decoding times, evaluating the RMSE together with the 99.8th and 100th (maximal error) percentiles of the root squared error distributions across 15,000 new uniformly sampled stimuli (*Figure 4e–f*). As suggested by the minimal decoding times, there is a clear trade-off between minimizing RMSE over longer decoding times and removing outliers, especially the maximal error, over shorter decoding times. *Figure 4—figure supplement 4* shows the time evolution for a few of these populations.

Additionally, to verify that the minimal decoding times are good predictors of the decoding time necessary to suppress large estimation errors, we compared the same error percentiles as in *Figure 4e–f* (i.e. the 99.8th and 100th percentiles) against the minimal decoding times, $T_{th}$, estimated in *Figure 4c*. For each population, we expect a strong reduction in the magnitude of the largest errors when the decoding time, $T$, is larger than the minimal decoding time, $T_{th}$. *Figure 5* shows a clear difference in large estimation errors between populations for which $T_{th} < T$ and populations with $T_{th} > T$ (circles to the left and right of the magenta lines in *Figure 5*, respectively). Thus, although only using the difference between MSE and Fisher information, our criterion on minimal decoding time still carries important information about the presence of large estimation errors.

To summarize, while periodic tuning curves provide lower estimation errors for long decoding times by minimizing local errors (*Figure 4c*, inset), a population of single-peaked tuning curves is faster at producing a statistically reliable signal by removing catastrophic errors (*Equation 9* and *Figure 4c*). Generalizing minimal decoding times to an arbitrary number of stimulus dimensions reveals that the minimal decoding time also depends on the stimulus dimensionality (*Figure 4c*, compare lines for $D = 1$ and $D = 2$). Interestingly, however, the approximation predicts that although minimal decoding time grows with increasing stimulus dimensionality, the minimal required spike count might be independent of stimulus dimensionality, at least for populations with integer spatial frequencies, that is, integer number of peaks (see *Equation A5.4*). The populations simulated here have non-integer spatial frequencies. However, the trend of changes in the mean spike count is still just slightly below 1 (indicating that slightly fewer spikes across the population are needed with increasing $D$, see *Figure 4—figure supplement 5*). Thus, as the average firing rate decreases with the number of encoded features

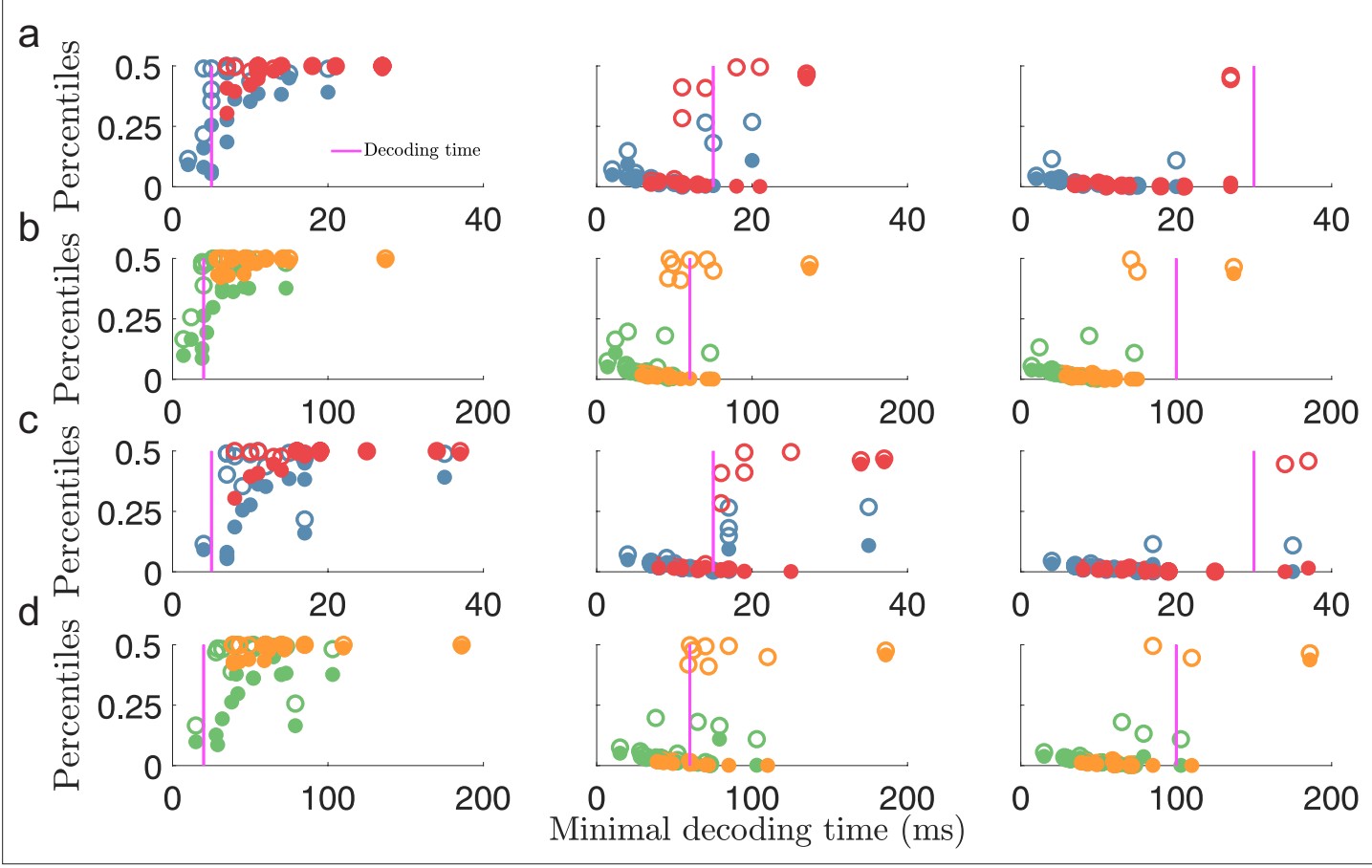

**Figure 5.** Minimal decoding time predicts the removal of large estimation errors. (**a**) The 99.8th percentile (filled circles) and the maximal error (i.e., 100th percentile, open circles) of the root squared error distributions for $D = 1$ against the estimated minimal decoding time for the corresponding populations ($\alpha = 2$) for various choices of decoding time, $T$ (indicted by the vertical magenta lines). (**b**) same as (**a**) but for $D = 2$. (**c-d**) Same as for (**a-b**) but for $\alpha = 1.2$. Note that the plots (**a**) and (**c**), or (**b**) and (**d**), illustrate the same percentile data only remapped on the x-axis by the different minimal decoding times from the different threshold factors $\alpha$. Color code: same as in **Figure 4**.

$D$ (**Figure 4b**, right column), the increase in minimal decoding time for stimuli of higher dimensionality can be primarily explained by requiring a longer time to accumulate the sufficient number of spikes across the population. Lastly, to rule out that the differences in minimal decoding time cannot be explained by the periodicity of the tuning curves not aligning to that of the stimulus, we also simulated populations with different combinations of integer peaks (**Figure 4—figure supplement 6**). Again, the same phenomenon is observed: periodic tuning curves increase the required decoding time to remove catastrophic errors. This also highlights that the approximation of minimal decoding time does not require the spatial periods to be related by a scale factor, $c$.

## Effect of ongoing activity

Many cortical areas exhibit ongoing activity, that is, activity that is not stimulus-specific (**Snodderly and Gur, 1995**; **Barth and Poulet, 2012**). Thus, it is important to understand the impact of ongoing activity on the minimal decoding time, too. Unfortunately, because our approximation of the minimal decoding times did not include ongoing activity, we relied on simulations to study the effect of such non-specific activity.

When including independent ongoing (background) activity at 2 spikes/s to all neurons for the same populations as in **Figure 4**, minimal decoding times were elevated across all populations (**Figure 6**). Furthermore, the minimal decoding time increased faster with decreasing $c$ in the presence of ongoing activity compared to the case without ongoing activity (ratios of fitted regressors $K_1(b = 2)/K_1(b = 0)$ using **Equation 47** were approximately 1.69 and 1.72 for $D = 1$ and $D = 2$, respectively). Similar results

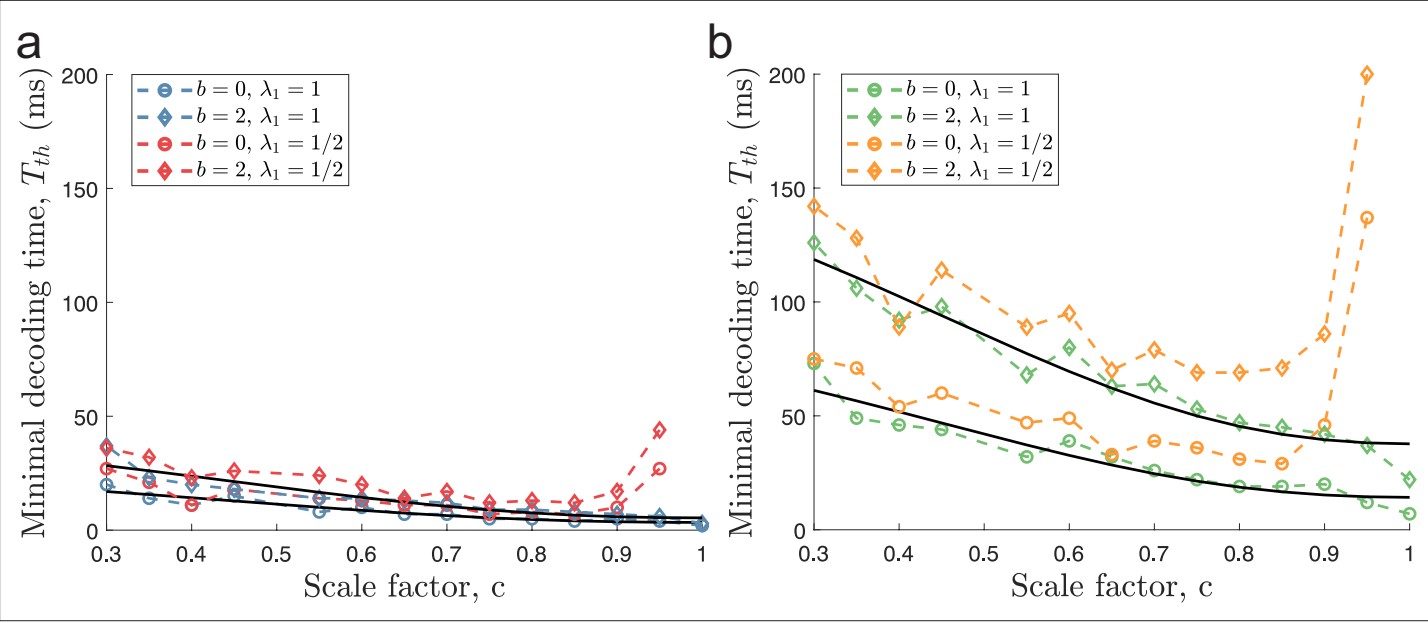

**Figure 6.** Ongoing activity increases minimal decoding time. (**a**) The case of encoding a one-dimensional stimulus ($D = 1$) with or without ongoing activity at 2 sp/s (diamond and circle shapes, respectively). (**b**) The case of a two-dimensional stimulus ($D = 2$) under the same conditions as for (**a**). In both conditions, ongoing activity increases the time required for all populations to produce reliable signals, but the effect is strongest for $c \ll 1$. The parameters used in the simulations are: population size $N = 600$, number of modules $L = 5$, scale factors $c = 0.3 - 1$, width parameter $w = 0.3$, average evoked firing rate $\overline{f_{stim}} = 20 \exp(-D/w) B_0(1/w)^D$ sp/s, ongoing activity $b = 2$ sp/s, and threshold factor $\alpha = 2$.

The online version of this article includes the following figure supplement(s) for figure 6:

**Figure supplement 1.** Minimal decoding times using a different threshold factor.

**Figure supplement 2.** Minimal decoding times based on one-sided KS-test.

are found using $\alpha = 1.2$ (*Figure 6—figure supplement 1*) or the alternative criterion on minimal decoding time based on one-sided KS-tests described earlier (*Figure 6—figure supplement 2*). Thus, ongoing activity can have a substantial impact on the time required to produce reliable signals. *Figure 6* suggests that areas with ongoing activity are less suited for periodic tuning curves. Especially, the combination of multidimensional stimuli and ongoing activity leads to much longer minimal decoding times for tuning curves with small spatial periods ($c \ll 1$). For example, when encoding a two-dimensional stimulus, only the populations with $\lambda_1 = 1$, $c = 1$ and $\lambda_1 = 1$, $c = 0.95$ could remove catastrophic errors in less than 40ms when ongoing activity at 2 sp/s was present. Thus, the ability to produce reliable signals at high speeds severely deteriorates for periodic tuning curves in the presence of non-specific ongoing activity.

This result has an intuitive explanation. The amount of catastrophic errors depends on the probability that the trial variability reshapes the neural activity to resemble the possible activities for a distinct stimulus condition (see *Figure 1a*). From the analysis presented above, periodic tuning curves have been suggested to be more susceptible to such errors. Adding ongoing activity does not reshape the stimulus-evoked parts of the tuning curves but only increases the trial-by-trial variability. Thus, by this reasoning, it is not surprising that the systems which are already more susceptible also are even more negatively affected by the increased variability induced by ongoing activity. The importance of *Figure 6* is that even ongoing activity as low as 2 sp/s can have a clearly visible effect on minimal decoding time.

## Implications for a simple spiking neural network with sub-optimal readout

Until this point, the arguments about minimal decoding time have relied on rate-based tuning curves encoding static stimuli. To extend beyond static stimuli and to exemplify the role of decoding time for spiking neurons, we simulated simple two-layer feed-forward spiking neural networks to decode

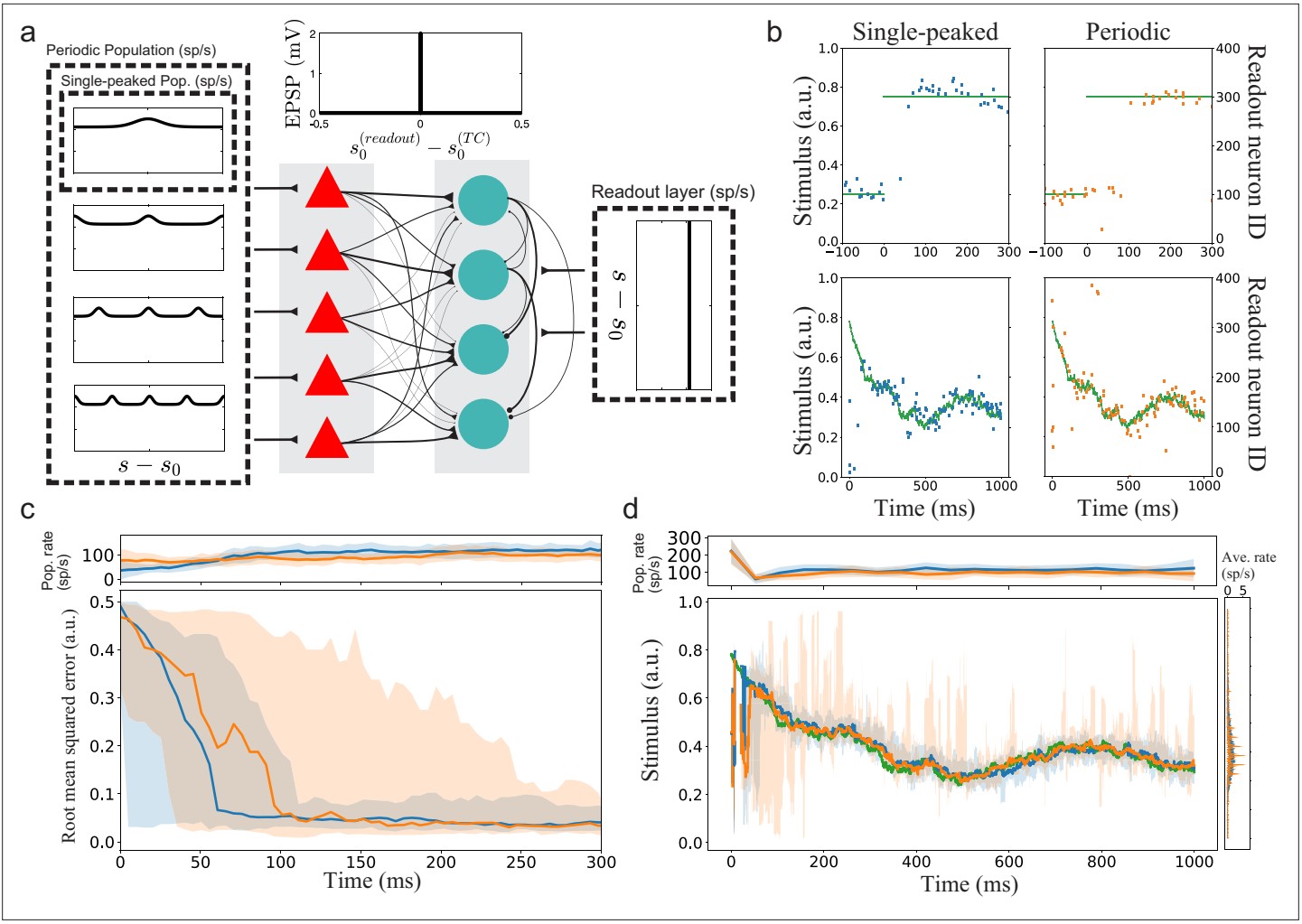

**Figure 7.** Implications for a simple spiking neural network with suboptimal readout. (**a**) Illustration of the spiking neural networks (SNNs). (**b**) Example of single trials. Top row: Two example trials for step-like change in stimulus (green line). The left and right plots show the readout activity for the single-peaked (blue) and periodic SSNs (orange), respectively. Note that the variance around true stimulus is larger for the single-peaked SNN (i.e. larger local errors) but that there are fewer very large errors than for the periodic SNN. Bottom row: Same as for the top row but with a continuously time-varying stimulus. (**c**) Bottom: The median RMSE (thick lines) over all trials in a sliding window (length 50ms) for the single-peaked (blue) and periodic (orange) SNNs. The shadings correspond to the regions between the 5th and 95th percentiles. Top: The instantaneous population firing rates of the readout layers and the standard deviations (same color code as in the bottom panel). (**d**) Bottom left: The median estimated stimulus across trials in a sliding window (length 10ms) for the single-peaked (blue) and periodic (orange) SNNs. Shaded areas again correspond to the regions between the 5th and 95th percentiles. The true stimulus is shown in green. Bottom right: the average firing rate of each neuron, arranged according to the preferred stimulus condition. Top: The instantaneous population firing rates of the readout layers and the standard deviations. See Materials and methods for simulation details and **Table 1**, **Table 2**, **Table 3** for all parameters used in the simulation.

time-varying stimulus signals. The first layer ($N_1 = 500$) corresponds to the tuning curves (without connections between the simulated neurons). The stimulus-specific tuning of the Poissonian inputs to these neurons is either fully single-peaked, creating a population of single-peaked tuning curves, or periodic with different spatial periods, creating a population of periodic tuning curves (**Figure 7a**, see Materials and methods for details). The second layer instead acts as a readout layer ($N_2 = 400$, allowing a weak convergence of inputs from the first layer). This layer receives both stimulus-specific excitatory input from the first layer and external non-specific Poissonian excitation (corresponding to background activity). The connection strength between the first and second layers depends on the difference in preferred stimulus conditions between the pre- and post-synaptic neurons. Such connectivity could, for example, be obtained by unsupervised Hebbian learning. Because the tuning curves in the first layer can be periodic, they can also connect strongly to several readout neurons.

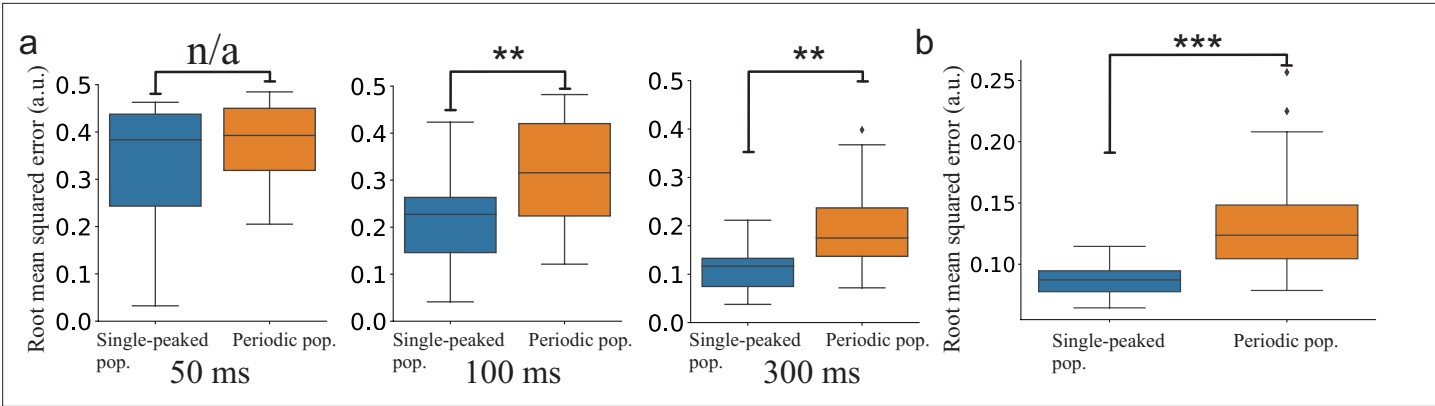

**Figure 8.** Statistical comparison of the SNN models. (**a**) Step-like change: Comparison between the distributions of accumulated RMSEs at different decoding times ($p = 0.4$, $9.0 \cdot 10^{-4}$, and $8.7 \cdot 10^{-5}$, respectively). (**b**) OU-stimulus: The distributions of RMSE across trials for the two SNNs ($p = 4.3 \cdot 10^{-8}$). All statistical comparisons in (**a**) and (**b**) were based on two-sample Kolmogorov–Smirnov (KS) tests using 30 trials per network.

We introduced lateral inhibition among the readout neurons (without explicitly modeling inhibitory neurons) to create a winner-take-all style of dynamics. The readout neurons with large differences in preferred stimulus inhibit each other more strongly. Decoding is assumed to be instantaneous and based on the preferred stimulus condition of the spiking neuron in the readout layer. However, to compare the readouts, we averaged the stimulus estimates in sliding windows.

We tested two different types of time-varying stimuli: (1) a step-like change from $s = 0.25$ to $s = 0.75$ (*Figure 7b* top row, green trace) and (2) a continuously time-varying stimulus drawn from an Ornstein–Uhlenbeck process (*Figure 7b* bottom row, green trace; see Materials and methods). In the case of a step-like stimulus change, the readout layer for the single-peaked population required a shorter time to switch states than the periodic network (*Figure 7c*). The shorter switching time is consistent with the hypothesis that single-peaked tuning curves have shorter minimal decoding times than periodic tuning curves. In these simulations, the difference is mainly due to some neurons in the first layer of the periodic network responding both before and after the step change. Thus, the correct readout neurons (after the change) must compensate for the hyper-polarization built up before the change and the continuing inhibitory input from the previously correct readout neurons (which still get excitatory inputs). Note that there are only minor differences in the population firing rates between the readout layers, confirming that this is not a consequence of different excitation levels but rather of the structures of excitation.

The continuously time-varying stimulus could be tracked well by both networks. However, averaging across trials shows that SNNs with periodic tuning curves have larger sporadic fluctuations (*Figure 7d*). This suggests that decoding with periodic tuning curves has difficulties in accurately estimating the stimulus without causing sudden, brief periods of large errors. To make a statistical comparison between the populations, we investigated the distributions of root mean squared error (RMSE) across trials. In both stimulus cases, there is a clear difference between the network with single-peaked tuning curves and the network with periodic ones. For the step-like change in stimulus condition, a significant difference in RMSE arises roughly 100 ms after the stimulus change (*Figure 8a*). For the time-varying stimulus, using single-peaked tuning curves also results in significantly lower RMSE compared to a population of periodic tuning curves (*Figure 8b*, RMSE calculated across the entire trial).

## Discussion

Several studies have suggested that periodic tuning creates an unparalleled precise neural code by minimizing local errors (*Sreenivasan and Fiete, 2011*; *Mathis et al., 2012*; *Wei et al., 2015*; *Malerba et al., 2022*). Nevertheless, despite this advantage of periodic tuning, single-peaked tuning curves are widespread in early sensory areas and especially in the early visual system. There is a long history of studying information representation using rate-based tuning curves. Still, the effect of spatial periodicity and catastrophic errors on the required decoding time has not been addressed. Here, we

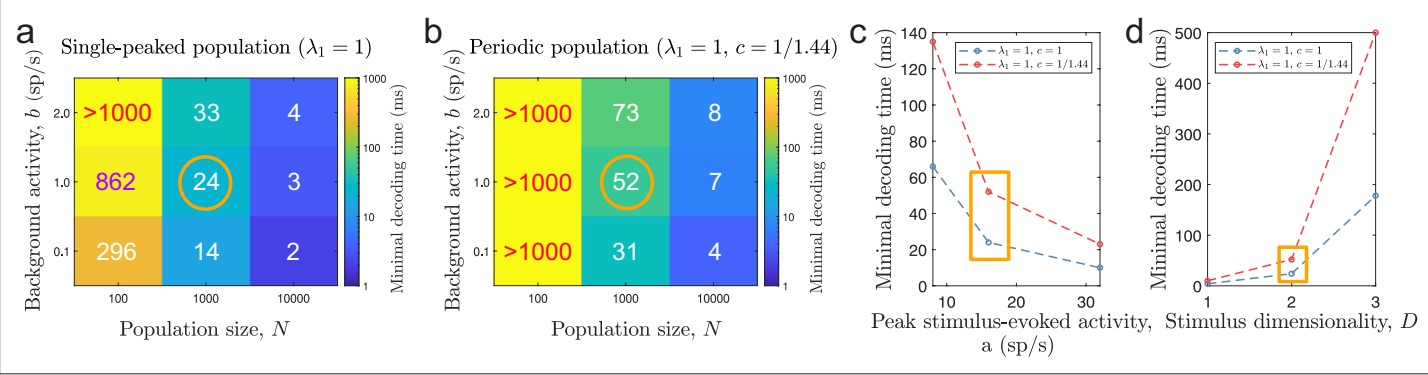

**Figure 9.** Minimal decoding time for various tuning and stimulus parameters. (**a-b**) Minimal decoding time for different combinations of population sizes ($N$) and levels of ongoing background activity ($b$) for the single-peaked population (**a**) and the periodic population (**b**). (**c**) Minimal decoding time as a function of average stimulus-evoked firing rate (x-axis re-scaled to the corresponding peak amplitude, $a$, for single-peaked tuning curves for easier interpretation). The corresponding amplitudes are $a = 8, 16$, and $32$ sp/s, respectively. (**d**) Minimal decoding time as a function of stimulus dimensionality. Unless indicated on the axes, the parameters are set according to the orange circles and rectangles in (**a-d**). Auxiliary parameters: number of modules $L = 5$, width parameter $w = 0.3$, and threshold factor $\alpha = 1.2$.

showed that the possibility of catastrophic estimation errors (*Figure 1a*) introduces the possibility that different shapes of tuning curves can have different minimal decoding times.

The emerging question is whether there is a trade-off between the accuracy of a neural code and the minimal required decoding time for single-peaked and periodic tuning. The answer is yes. We found that minimal decoding time increased with decreasing spatial periods of the tuning curves (*Figure 4c*), suggesting a trade-off between accuracy and speed for populations of tuning curves. The differences in minimal decoding time cannot be explained by the periodicity of the tuning curves not aligning to the stimulus domain, as the same holds comparing populations with integer number of peaks (*Figure 4—figure supplement 6*). Furthermore, our results remained unchanged when we discarded any decoded stimuli which needed the mod 1 operation to lie within the stimulus domain $[0, 1)^D$, thus ruling out any possible distortion effect of the periodic stimulus and decoding approach. In addition, we show that our results are valid for a range of population sizes (*Figure 9a-b*), ongoing (*Figure 9a-b*) and evoked activities (*Figure 9c*), and stimulus dimensions (*Figure 9d*). We used the more conservative threshold factor on MSE, $\alpha = 1.2$, to capture all the nuances w.r.t. the level of ongoing activity even for large population sizes. In simulated networks with spiking neurons, we showed that the use of periodic tuning curves increased the chances of large estimation errors, leading to longer times before switching 'states' (*Figure 7c*) and difficulties tracking a time-varying, one-dimensional stimulus (*Figure 7d*).

Experimental data suggest that decoding times can be very short, of the order of tens of milliseconds, reflecting that a considerable part of the information contained in firing rates over long periods is also present in short sample periods (*Tovée et al., 1993*). Additionally, the first few spikes have been shown to carry significant amounts of task information in both visual (*Resulaj et al., 2018*), olfactory (*Resulaj and Rinberg, 2015*), and somatosensory areas (*Panzeri et al., 2001*; *Petersen et al., 2001*). As the tuning curves in this study all have equal average firing rates, we can reinterpret the minimal decoding time in terms of the prominence of the first spikes. In our simulations, tens of spikes carry enough information to produce a reliable stimulus estimate free of catastrophic errors (*Figure 4—figure supplement 5*). As with decoding time, single-peaked tuning curves also need fewer spikes to produce reliable signals. Thus, the speed-accuracy trade-off can be reinterpreted as a trade-off between being accurate and efficient.

The notion of a speed-accuracy trade-off is further strengthened when considering high-dimensional stimuli that demand longer minimal decoding times. Natural stimuli typically have higher dimensionality than those used in animal experiments. Many sensory neurons are tuned to multiple features of the external stimulus, creating mixed selectivity of features (e.g. *Garg et al., 2019*). For neurons responding to task-related variables, mixed selectivity has been shown to enable linear separability and to improve discriminability (*Rigotti et al., 2013*; *Fusi et al., 2016*; *Johnston et al., 2020*). For continuous stimulus estimations, mixed selectivity has also been proposed to decrease MSE when

decoding time is limited (*Finkelstein et al., 2018*). However, to remove catastrophic errors, which, as we have argued, is not necessarily synonymous with lower MSE, the exponential increase in minimal decoding time could easily lead to very long decoding times. Thus, minimal decoding time should set a bound on the number of features a population can jointly encode reliably. In addition, neurons in sensory areas often exhibit a degree of non-specific activity (*Snodderly and Gur, 1995*; *Barth and Poulet, 2012*). Introducing ongoing activity to the populations in our simulations further amplified the differences in minimal decoding times (*Figure 6*). Thus, for jointly encoded stimuli, especially in areas with high degrees of ongoing activity, a population of single-peaked tuning curves might be the optimal encoding strategy for rapid and reliable communication.

We note that these results might extend beyond the visual areas, too. Although this study focused on tuning curves encoding continuous variables, catastrophic errors can also occur in systems with discrete tuning curves. Sensory stimuli can be fast-varying (or even discontinuous), and large errors can potentially harm the animal. Therefore, constraints on decoding time are likely important for any early sensory area. In addition, hippocampal place cells involved in spatial navigation (*O'Keefe and Dostrovsky, 1971*; *Wilson and McNaughton, 1993*) are known for their single-peaked tuning. The interesting observation in this context is that place cells produce reliable signals faster than their input signals from the medial entorhinal cortex with a combination of single- and multi-peaked tuning (*Cholvin et al., 2021*). On the other hand, for sufficiently slow-varying stimuli, a periodic population can be used together with error correction to remove catastrophic errors (*Sreenivasan and Fiete, 2011*). Furthermore, for non-periodic stimuli with large domains, the combinatorial nature of periodic tuning curves can create unique stimulus representations far exceeding the spatial periods of the tuning curves (*Fiete et al., 2008*). Thus, periodic tuning curves are ideal for representing space, where the stimulus domain can be vast, and the change in position is constrained by the speed of movement. Interestingly, when faced with very large arenas, place cells can also exhibit multi-peaked tuning (*Eliav et al., 2021*).

To summarize, we provide normative arguments for the single-peaked tuning of early visual areas. Rapid decoding of stimulus is crucial for the survival of the animals. Consistent with this, animals and humans can process sensory information at impressive speeds. For example, the human brain can generate differentiating event-related potentials to go/no-go categorization tasks using novel complex visual stimuli in as little as 150 ms (*Thorpe et al., 1996*). These 'decoding' times do not decrease for highly familiar objects, suggesting that the speed of visual processing cannot be reduced by learning (*Fabre-Thorpe et al., 2001*). Given constraints on low latency communication, it is crucial that each population can quickly produce a reliable signal. In this regard, single-peaked tuning curves are superior to periodic ones. The fact that early visual areas exhibit ongoing activity and encode multi-dimensional stimuli further strengthens the relevance of the differences in minimal decoding time.

To conclude, our work highlights that minimum decoding time is an important attribute and should be considered while evaluating candidate neural codes. Our analysis suggests that decoding of high-dimensional stimuli can be prohibitively slow with rate-based tuning curves. Experimental data on the representation of high-dimensional stimuli is rather scant as relatively low-dimensional stimuli are typically used in experiments (e.g. oriented bars). Our work gives a compelling reason to understand whether and how biological brains can reliably encode high-dimensional stimuli at behaviorally relevant time scales.

# Materials and methods
## Minimal decoding times - simulation protocols

To study the dependence of decoding time $T$ on MSE for populations with different distributions of spatial frequencies, we simulated populations of synthetic tuning curves (*Equation 1*). The stimulus was circular with a $[0, 1)^D$ range. The preferred stimulus conditions $\mathbf{s}'$ were sampled independently from a random uniform distribution over $[0, 1)^D$ (independently and uniformly for each stimulus dimension). To ensure equal comparison, the preferred locations $\mathbf{s}'$ were shared across all populations. Each neuron's amplitude, $a_i$, was tuned according to *Equation A1.5* to ensure an equal average firing rate across the stimulus domain for all neurons. In each trial, a stimulus $\mathbf{s} \in [0, 1)^D$ was also independently

sampled from a uniform distribution over $[0, 1)^D$. The spike count for each neuron was then sampled according to *Equation 3*.

Minimal decoding time was defined as the shortest time for which the neural population approximately reaches the Cramér-Rao bound. To estimate the reaction time in simulations, we incrementally increased the decoding time $T$ (using 1 ms increments, starting at $T = 1$ ms) until

$$\overline{MSE(T, \lambda)} \leq \alpha \cdot \overline{\operatorname{diag}(J(T, \lambda)^{-1})}. \tag{10}$$

As the ML estimator is asymptotically efficient (attaining the Cramér-Rao bound in the limit of infinite data), the threshold factor, $\alpha$, in *Equation 10* was added as a relaxation (see figure captions for choices of $\alpha$). Note that the mean bars on the left- and right-hand sides of *Equation 10* refer to the means across stimulus dimensions (for multi-dimensional stimuli) and that $\operatorname{diag}(\cdot)$ refers to taking the diagonal elements from the inverse of the Fisher information matrix, $J(T, \lambda)^{-1}$. For a given decoding time $T$, the estimation of MSE was done by repeatedly sampling random stimulus conditions (from a uniform distribution), sampling a noisy response to the stimulus (Poisson distributed spike counts), and then applying maximum likelihood estimation (see next section 'Implementation of maximum likelihood estimator' for details on implementation). In *Figures 3 and 9*, 15000 stimulus conditions were drawn for each $T$, and in *Figure 4*, stimulus conditions were repeatedly drawn until the two first non-zero digits of the MSE were stable for 1000 consecutive trials. However, in controls not presented here, we could not see any significant difference between these two sampling approaches. Because the Fisher information matrix $J$ was analytically estimated only in the special case without ongoing activity, it was approximated in simulations by the element-wise average across 10,000 randomly sampled stimulus conditions (also uniformly distributed), where each element was calculated according to *Equation A2.12* or *Equation A2.13* given a random stimulus trial.

## Implementation of maximum likelihood estimator

Given some noisy neural responses, $\mathbf{r}$, the maximum likelihood estimator (MLE) chooses the stimulus condition which maximizes the likelihood function, $\hat{\mathbf{s}}_{ML} = \arg\max_{\mathbf{s}} \mathcal{L}(\mathbf{r}, \mathbf{s}) = \arg\max_{\mathbf{s}} \prod_{i=1}^{N} p(r_i|\mathbf{s})$. A common approach is to instead search for the maximum of the log-likelihood function (the logarithm is a monotonic function and therefore preserves any maxima/minima). The stimulus-dependent terms of the log-likelihood can then be expressed as

$$\log p(\mathbf{r}|\mathbf{s}) \propto \mathcal{V}(\mathbf{r}; \mathbf{s}) = \sum_{i=1}^{N} r_i \log(T f_i(\mathbf{s})) - T f_i(\mathbf{s}). \tag{11}$$

Unfortunately, the log-likelihood function is not guaranteed to be concave, and finding the stimulus condition $\hat{\mathbf{s}}_{ML}$ which maximizes the log-likelihood function is not trivial (a non-convex optimization problem). To overcome this difficulty, we combined grid-search with the Nelder–Mead method, an unconstrained non-linear program solver (implemented using MATLAB's built-in function *fminsearch*, https://www.mathworks.com/help/matlab/ref/fminsearch.html).

Grid search was used to find a small set of starting points with large log-likelihood values. To do so, we sampled 100 random stimulus conditions within the stimulus interval $[0, 1)^D$ and selected the four stimulus conditions with the largest log-likelihood values. The true stimulus condition, $\mathbf{s}_{true}$, was always added to the set of starting points regardless of the log-likelihood value of that condition (yielding a total of 5 starting points).

Then the Nelder–Mead method was used with these starting points to find a set of 5 (possibly local) maxima. The stimulus was decoded as the stimulus, $\hat{\mathbf{s}}$, yielding the largest log-likelihood of the 5 maxima. As we always included the true stimulus condition in the Nelder-Mead search, this approach should not overestimate the amount of threshold distortion but can potentially miss some global estimation errors instead. Finally, as the Nelder–Mead method is unconstrained but the stimulus domain periodic, the output of the maximum likelihood decoder was transformed into the stimulus interval $[0, 1)^D$ by applying the mod 1 operation on each stimulus dimension,

$$\hat{\mathbf{s}}_{ML} = \hat{\mathbf{s}} \ (\operatorname{mod} \ 1). \tag{12}$$

**Table 1.** Parameters and parameter values for O-U stimulus.

| Parameters | Parameter values |
|---|---|
| $\tau_s$ | 0.5 (s) |
| $\sigma_s$ | 0.1 |

Given an estimated stimulus, $\hat{s}_{ML}$, the error was then evaluated along each stimulus dimension independently, taking into account the periodic boundary,

$$\epsilon_i^2 = \min\left[(s_i - \hat{s}_{ML,i})^2, (s_i - \hat{s}_{ML,i} + 1)^2, (s_i - \hat{s}_{ML,i} - 1)^2\right] \tag{13}$$

for $i \in \{1, ..., D\}$.

Lastly, to rule out that the estimates before the mod-operation, $\hat{\mathbf{s}}$, outside of the stimulus domain $[0, 1)^D$ did not influence the results, we also discarded these samples, but this produced similar results.

## Spiking network model

### Stimuli

As in the previous simulations, we assumed that the stimulus domain was a circular stimulus defined between $[0, 1)$. We simulated the responses to two different types of stimuli, (1) a step-like change in stimulus condition from $s = 0.25$ to $s = 0.75$ and (2) a stimulus drawn from a modified Ornstein–Uhlenbeck process

$$\frac{ds_I}{dt} = -\frac{s_I}{\tau_s} + \sqrt{\frac{2\sigma_s^2}{\tau_s}}\xi_s \quad (\text{mod } 1). \tag{14}$$

For parameter values, see **Table 1**.

### Network model

The spiking networks were implemented as two-layer, feed-forward networks using LIF neurons with (Dirac) delta synapses. The dynamics of the membrane potential for the neurons in the first layer were described by

$$\frac{dV_i}{dt} = -\frac{V_i - V_{rest}}{\tau_{mem}} + \sum_k J_E \delta(t - t_k), \tag{15}$$

where $V_i$ is the voltage of neuron $i$, $\tau_{mem}$ the membrane time constant, $t_k$ the timing of the $k$ th input spike to neuron $i$, and $J_E$ the induced EPSP. The neurons in the first layer were constructed to correspond to either single-peaked or periodic tuning curves. Two networks were tested, one network where the first layer corresponds to single-peaked tuning curves and a second network corresponding to periodic tuning curves (with $L = 4$ modules). For each neuron $i$ in module $j$ in the first layer, the input was drawn from independent Poisson point processes with stimulus-dependent rates $f_i^{(j)}(s(t))$

$$f_i^{(j)}(s(t)) = a\exp\left(\frac{1}{w}\left(\cos(\frac{2\pi}{\lambda_j}(s(t) - s_i^{(j)})) - 1\right)\right) + b. \tag{16}$$

Here, the constants $a$ and $b$ were chosen such that the baseline firing rate was slightly above zero and the maximal firing rate was slightly below 20 sp/s (see Table 3 for all network-related parameter values). Because of the choice of $\lambda_j$, the modulation strengths of the inputs were such that the average

**Table 2.** Parameters and parameter values for LIF neurons.

| Parameters | Parameter values |
|---|---|
| Membrane time constant, $\tau_{memb}$ (ms) | 20 |
| Threshold memb. potential, $V_{th}$ (mV) | 20 |
| Reset memb. potential (mV) | 10 |
| Resting potential, $V_0$ (mV) | 0 |
| Refractory period, $\tau_{rp}$ (ms) | 2 |

**Table 3.** Spiking network parameters and parameter values.

| Parameters | Parameter values |
| --- | --- |
| Number of neurons 1st layer, $N_1$ | 500 |
| Number of neurons 2nd layer, $N_2$ | 400 |
| Maximal stimulus-evoked input rate, $a$ (sp/s) | 750 |
| Baseline input rate, $b$ (sp/s) | 4250 |
| Spatial periods, $\lambda_j$ | [1] or [1,2,3,4] |
| Width parameter, $w$ | 0.3 |
| Width parameter (readout layer), $w_{ro}$ | $\frac{(\pi/N_2)^2}{2\log(2)}$ |
| Input EPSP (1st layer), $J_E$ (mV) | 0.2 |
| Maximal EPSP (2nd layer), $J_{EE}$ (mV) | 2 |
| Maximal IPSP (2nd layer), $J_{II}$ (mV) | 2 |
| Synaptic delays, $d$ (ms) | 1.5 |

input to each neuron was equal. For each module in the first layer, the preferred locations $s_i^{(j)}$ were equidistantly placed across $[0, \lambda_j)$.

Similarly, for the second layer, the membrane potential was described by

$$\frac{dV_i}{dt} = -\frac{V_i - V_{rest}}{\tau_{mem}} + \sum_{j\in[1,...,N_1]}\sum_k J_{EE}(\Delta_{i,j})\delta(t - t_k^{(j)} - d) + \sum_{j\in[1,...,N_2]}\sum_k J_I(\Delta_{i,j})\delta(t - t_k^{(j)} - d) + \sum_k J_E\delta(t - t_k),$$

(17)

where $J_{EE}(\Delta_{i,j})$ and $J_I(\Delta_{i,j})$ are synapse-specific EPSPs/IPSPs which depends on the difference in preferred tuning $\Delta_{i,j}$ between the pre- and post-synaptic neurons (see *Equation 18*), $t_k^{(j)}$ the timing of the $k$ th spike of pre-synaptic neuron $j$, and $d$ the delay (see *Table 2*, *Table 3* for parameter values). The neurons in the second layer were only tuned to a single preferred stimulus location each, equidistantly placed across $[0, 1)$. Whenever a spike occurred in the first layer, it elicited EPSPs with a delay of 1.5 ms in all neurons in the second layer. The size of the EPSPs depended on the difference in preferred tuning, $\Delta_{i,j}$, between the pre- and post-synaptic neurons

$$J_{EE}(\Delta_{i,j}) = \exp(\tfrac{1}{w_{ro}}(\cos(2\pi\Delta_{i,j}) - 1))J_{EE}.$$

(18)

Here, $J_{EE}$ determines the maximal EPSP (mV), and the constant $w_{ro}$ was chosen such that the full width at half maximum of the EPSP kernels tiled the stimulus domain without overlap. Note that for periodically tuned neurons in the first layer (i.e. with multiple preferred locations), the $\Delta_{i,j}$ were determined by the smallest difference in preferred tuning across the multiple preferred locations.

As for the excitatory neurons in the first layer, whenever a spike occurred in the second layer, it elicited IPSPs with a delay of 1.5 ms in all other neurons in the second layer. Again, the size of the IPSPs depended on the difference in preferred tuning, $\Delta_{i,j}$ between the two neurons, but this time according to

$$J_I(\Delta_{i,j}) = -\left| \sin(\pi\Delta_{i,j}) \right| J_I.$$

(19)

Thus, the range of inhibition was much broader compared to the excitation.

## Evaluating decoding performance

We assumed that the decoder was instantaneously based on the neuron index of the firing neuron in the readout layer. Let $\Phi(t_k)$ denote a function that provides the index of the neuron firing at time $t_k$. Given the equidistant distribution of preferred locations for the readout neurons, the stimulus is instantaneously decoded by mapping the neuron identity to the interval $[0, 1]$

$$\hat{s}(t_k) = \frac{\Phi(t_k)}{N_2}, \tag{20}$$

where $N_2$ is the number of neurons in the readout layer. For both stimulus cases, the decoding performance was evaluated using (1) the distribution of RMSE (*Figure 7d*) or estimated stimulus conditions (*Figure 7e*) in a sliding window or (2) the distributions of accumulated RMSE (*Figure 8*).

## Parameters

### Simulation tools

All the simulations were done using code written in MATLAB and Python (using Brian2 simulator *Stimberg et al., 2019*). The simulation code is available at https://github.com/movitzle/Short_Decoding_Time (copy archived at *Lenninger, 2023*).

## Approximating minimal decoding time in two-module systems

To gain an understanding of the interaction between two modules with different spatial periods, consider the likelihood function as a product of the likelihood functions of the two modules individually

$$p(\mathbf{r}|s) = Q_1(s)Q_2(s). \tag{21}$$

Using the Laplace approximation, each of these functions can be approximated as a periodic sum of Gaussians (*Wei et al., 2015*). Assuming that each module becomes efficient before the joint likelihood, we only focus on the largest, periodically occurring, peaks

$$Q_i(s) \approx \hat{Q}_i(\mathbf{r}^{(i)}|s) = A_i \sum_{n_i=-K_i}^{K_i} \exp\left( -\frac{\Sigma_i}{2}\left(s - (s_{ML}^{(i)} + n_i\lambda_i)\right)^2 \right), \tag{22}$$

where $\mathbf{r}^{(i)}$ denotes the activity pattern of module $i$, $s_{ML}^{(i)}$ the peak closest to the true stimulus condition, $s_0$, and $K_i$ is large enough for $\hat{Q}_i(r^{(i)}|s)$ to cover the entire stimulus range $[0, 1)$. The approximation can be seen as 'rolling out' the stimulus domain from $[0, 1)$ to $\mathbb{R}$. Therefore, to neglect the impact of the stimulus boundary, we assume that the stimulus is in the middle of the stimulus domain and $K_1 = \lceil \frac{1}{2\lambda_1} \rceil$ and $K_2 = \lceil \frac{1}{2\lambda_2} \rceil$. Furthermore, assuming that each module is efficient, the width of the Gaussians can be approximated as

$$\Sigma_i \approx -\frac{d^2}{ds^2} \log Q_i(s) \approx J_i(s), \tag{23}$$

where $J_i(s) \approx J_i$ is the Fisher information of module $i$. The joint likelihood function can thus be approximated as

$$\begin{aligned} p(\mathbf{r}|s) \quad &\approx \hat{Q}_1(\mathbf{r}^{(1)}|s)\hat{Q}_2(\mathbf{r}^{(2)}|s) = \\ &= A_1A_2 \sum_{n_1=-K_1}^{K_1} \exp\left( -\frac{J_1}{2}\left(s - (s_{ML}^{(1)} + n_1\lambda_1)\right)^2 \right) \sum_{n_2=-K_2}^{K_2} \exp\left( -\frac{J_2}{2}\left(s - (s_{ML}^{(2)} + n_2\lambda_2)\right)^2 \right). \end{aligned} \tag{24}$$

As the likelihood functions depend on the particular realization of the spike counts, the distance between the modes of the respective likelihoods closest to the true stimulus condition $s_0$, $\delta_{0,0} = s_{ML}^{(1)} - s_{ML}^{(2)}$, is a random variable. Note that in the Results section, $\delta_{0,0}$ is simply referred to as $\delta$ for clarity.

The joint likelihood distribution $p(\mathbf{r}|s)$ has its maximal peak close to the true stimulus condition $s_0$ if $\delta_{0,0}$ is the smallest distance between any pairs of peaks of $Q_1$ and $Q_2$ (see *Equation A3.7* for details). Assuming that both modules provide efficient estimates, the distance $\delta_{0,0}$ can be approximated as a normally distributed random variable

$$\delta_{0,0} = s_{ML}^{(1)} - s_{ML}^{(2)} = (s_{ML}^{(1)} - s_0) - (s_{ML}^{(2)} - s_0) \sim \mathcal{N}\left(0, \frac{1}{T}(J_{1,norm}^{-1} + J_{2,norm}^{-1})\right), \tag{25}$$

where $J_{k,norm}$ refers to the time-normalized Fisher information of module $k$. Thus, as the decoding time $T$ increases, the variance of $\delta_{0,0}$ decreases. Hence, it is necessary for the decoding time $T$ to be large enough such that it is rare for $\delta_{0,0}$ not to be the smallest distance between any pair of peaks. Similarly, the distance between the other pair of peaks in $Q_1$ and $Q_2$ within the stimulus range becomes

$$\begin{aligned} \delta_{n_1,n_2} \quad &= (s_{ML}^{(1)} + n_1\lambda_1) - (s_{ML}^{(2)} + n_2\lambda_2) = \\ &= \delta_{0,0} + (n_1\lambda_1 - n_2\lambda_2), \end{aligned} \tag{26}$$

where $n_1 \in \{-K_1, ..., K_1\}$ and $n_2 \in \{-K_2, ..., K_2\}$ are indexing the different Gaussians as before. Thus, the threshold for catastrophic error is reached when there is another pair of modes with the same distance between them, that is,

$$\left|\delta_{0,0}\right| = \left|\delta_{n_1,n_2}\right| = \left|\delta_{0,0} + (n_1\lambda_1 - n_2\lambda_2)\right|, \qquad (27)$$

for some $n_1$ and $n_2$ belonging to the index sets as above. Thus, to avoid catastrophic errors, it is necessary that

$$\left|\delta_{0,0}\right| \leq \left|\delta_{0,0} + (n_1\lambda_1 - n_2\lambda_2)\right|, \qquad (28)$$

for all $n_1 \in \{-K_1, ..., K_1\}$ and $n_2 \in \{-K_2, ..., K_2\}$. By solving *Equation 28*, and taking into account that $\delta_{0,0}$ can be either positive or negative, we get the maximally allowed displacement

$$\delta^* = \min_{n_1,n_2:(n_1,n_2)\neq(0,0),n_1\in\{-K_1,...,K_1\},n_2\in\{-K_2,...,K_2\}} \frac{1}{2}\left|(n_1\lambda_1 - n_2\lambda_2)\right|. \qquad (29)$$

Note that for $\lambda_1 = 1$, all $n_1$ represent the same mode (but one full rotation 1 away). Thus, we limit the search such that $\lambda_1|n_1| < 1$ and $\lambda_2|n_2| < 1$. Assuming that the period of the second module is a scaling of the first module, $\lambda_2 = c\lambda_1$, the above equation becomes

$$\delta^* = \min_{n_1,n_2:(n_1,n_2)\neq(0,0)} \frac{1}{2}\left|\lambda_1(n_1 - n_2 c)\right|. \qquad (30)$$

Note that stimulus ambiguity can never be resolved if $\delta_{n_1,n_2} = \delta_{0,0}$ for some pair $(n_1, n_2) \neq (0,0)$, which is analogous to the condition in *Mathis et al., 2012*. To limit the probability of catastrophic estimation errors from the joint distribution to some small error probability $p_{error}$, the following should hold

$$Pr(|\delta_{0,0}| > \delta^*) < p_{error}. \qquad (31)$$

Because $\delta_{0,0} \sim \mathcal{N}(0, J_1^{-1} + J_2^{-1})$, we have

$$Pr(|\delta_{0,0}| > \delta^*) = 1 - \mathrm{erf}\left(\frac{\delta^*}{\sqrt{2}\sigma}\right) < p_{error}, \qquad (32)$$

where $\mathrm{erf}(\cdot)$ is the error-function and $\sigma = \sqrt{J_1^{-1} + J_2^{-1}}$. By rearranging the terms and using *Equation A2.8*, we can obtain a lower bound on the required decoding time

$$T_{th} > 2\left(\frac{\mathrm{erf}^{-1}(1-p_{error})}{\delta^*}\right)^2 \left(\frac{1}{J_{1,norm}} + \frac{1}{J_{2,norm}}\right), \qquad (33)$$

where $J_{i,norm}$ is the time-normalized Fisher information of module $i$. Note that $\delta^*$ can easily be found using an exhaustive search according to *Equation 29* or *Equation 30*.

## Approximating minimal decoding time

To approximate the order by which the population reaction time scales with the distribution of spatial periods and the stimulus dimensionality, we extended the approximation method introduced by *Xie, 2002*. The key part of the approximation method is to use a Taylor series to reason about which conditions must hold for the distribution of errors to be normally distributed with a covariance equal to the inverse of the Fisher information matrix. Note that this approximation assumes the existence of a unique solution to the maximum likelihood equations, thus, it does not apply to ambiguous neural codes (e.g. $c = 1/2, 1/3, 1/4, ...,$ etc.).

First, let's recollect the Taylor series with Lagrangian reminder for a general function g

$$g(x + \delta) = g(x) + g'(x)\delta + \frac{1}{2}g''(x^*)\delta^2, \qquad (34)$$

where $x^*$ is somewhere on the interval $[x, x + \delta]$. Thus, in the multivariate case, the derivative in the j:th direction of the log-likelihood function for stimulus condition $\hat{s}_{ML} = \hat{s}$ can be rewritten using a Taylor series with Lagrangian reminder as

$$\left.\frac{\partial}{\partial s_k} \log p(\mathbf{r}|\mathbf{s})\right|_{\mathbf{s}=\hat{\mathbf{s}}} = \left.\frac{\partial}{\partial s_k} \log p(\mathbf{r}|\mathbf{s})\right|_{\mathbf{s}=\mathbf{s}^\circ} + \sum_{l=1}^{D} \left.\frac{\partial^2}{\partial s_l \partial s_k} \log p(\mathbf{r}|\mathbf{s})\right|_{\mathbf{s}=\mathbf{s}^\circ} (\hat{s}_l - s_l^\circ) +$$
$$+ \frac{1}{2} \sum_{l=1}^{D} \sum_{m=1}^{D} \left.\frac{\partial^3}{\partial s_m \partial s_l \partial s_k} \log p(\mathbf{r}|\mathbf{s})\right|_{\mathbf{s}=\mathbf{s}^*} (\hat{s}_l - s_l^\circ)(\hat{s}_m - s_m^\circ),$$

(35)

for all $k \in \{1, ..., D\}$ where $\mathbf{s}^\circ$ is the true stimulus condition and $\mathbf{s}^*$ is a stimulus point between $\mathbf{s}^\circ$ and $\hat{\mathbf{s}}$.

If the estimated stimulus is close to the true stimulus, then the quadratic order terms are small. If so, the variance of $(\hat{\mathbf{s}} - \mathbf{s}^\circ)$ converges towards $\mathcal{N}(0, J^{-1})$ (in distribution), where $J$ is the Fisher information matrix (*Lehmann and Casella, 1998*). However, if the estimated stimulus is not close to the true stimulus, then the quadratic terms are not negligible. Therefore, when $T$ is sufficiently large, and the variance of the estimation follows the Cramér-Rao bound, the following should hold for all $k \in \{1, ..., D\}$

$$\left| \sum_{l=1}^{D} \left.\frac{\partial^2}{\partial s_l \partial s_k} \log p(\mathbf{r}|\mathbf{s})\right|_{\mathbf{s}=\mathbf{s}^\circ} (\hat{s}_l - s_l^\circ) \right| \gg \left| \frac{1}{2} \sum_{l=1}^{D} \sum_{m=1}^{D} \left.\frac{\partial^3}{\partial s_m \partial s_l \partial s_k} \log p(\mathbf{r}|\mathbf{s})\right|_{\mathbf{s}=\mathbf{s}^*} (\hat{s}_l - s_l^\circ)(\hat{s}_m - s_m^\circ) \right|.$$

(36)

In this regime, we make the following term-wise approximations

$$\left.\frac{\partial^2}{\partial s_l \partial s_k} \log p(\mathbf{r}|\mathbf{s})\right|_{\mathbf{s}=\mathbf{s}^\circ} \approx \mathbb{E}\left[ \left.\frac{\partial^2}{\partial s_l \partial s_k} \log p(\mathbf{r}|\mathbf{s})\right|_{\mathbf{s}=\mathbf{s}^\circ} \right] = -J_{k,l}(\mathbf{s}^\circ) = -J_{k,l},$$

(37)

and

$$\left.\frac{\partial^3}{\partial s_m \partial s_l \partial s_k} \log p(\mathbf{r}|\mathbf{s})\right|_{\mathbf{s}=\mathbf{s}^*} \approx \mathbb{E}\left[ \left.\frac{\partial^3}{\partial s_m \partial s_l \partial s_k} \log p(\mathbf{r}|\mathbf{s})\right|_{\mathbf{s}=\mathbf{s}^*} \right] = M_{k,l,m}(\mathbf{s}^*),$$

(38)

which gives

$$\left| \sum_{l=1}^{D} J_{k,l}(\hat{s}_l - s_l^\circ) \right| \gg \left| \frac{1}{2} \sum_{l=1}^{D} \sum_{m=1}^{D} M_{k,l,m}(\mathbf{s}^*)(\hat{s}_l - s_l^\circ)(\hat{s}_m - s_m^\circ) \right|.$$

(39)

Because $M_{k,l,m} \approx 0$ unless $k = l = m$ (see *Equation A4.3*, *Equation A4.4*, *Equation A4.5*), *Equation 39* simplifies to

$$\left| \sum_{l=1}^{D} J_{k,l}(\hat{s}_l - s_l^\circ) \right| \gg \left| \frac{1}{2} M_{k,k,k}(\mathbf{s}^*)(\hat{s}_k - s_k^\circ)^2 \right|.$$

(40)

Furthermore, because $J(\mathbf{s})$ is a diagonal matrix (see *Equation A2.18*), we have

$$\left| J_{k,k}(\hat{s}_k - s_k^\circ) \right| \gg \left| \frac{1}{2} M_{k,k,k}(\mathbf{s}^*)(\hat{s}_k - s_k^\circ)^2 \right|.$$

(41)

Next, by taking the square of the absolute values, we obtain

$$J_{k,k}^2(\hat{s}_k - s_k^\circ)^2 \gg \frac{1}{4} M_{k,k,k}^2(\mathbf{s}^*)\left( (\hat{s}_k - s_k^\circ)^2 \right)^2.$$

(42)

Because we assumed that $N$ and $T$ are sufficiently large to meet the Cramér-Rao bound, we have that

$$(\hat{s}_k - s_k^\circ)(\hat{s}_l - s_l^\circ) \sim \{\bar{J}^{-1}\}_{k,l}.$$

(43)

Inserting *Equation 43* into *Equation 42* gives

$$J_{k,k}^2 \{J^{-1}\}_{k,k} \gg \frac{1}{4} M_{k,k,k}^2(\mathbf{s}^*)\left( \{J^{-1}\}_{k,k} \right)^2,$$

(44)

or, equivalently,

$$1 \gg \frac{1}{4} M_{k,k,k}^2(\mathbf{s}^*)\{J^{-1}\}_{k,k}^3 = \frac{1}{4} \frac{M_{k,k,k}^2(\mathbf{s}^*)}{\{J\}_{k,k}^3}.$$

(45)

By approximating the term $M_{k,k,k}(\mathbf{s}^*)$ with an upper bound $M^*$ (see *Equation A4.10*) and using the expression for Fisher information (*Equation A2.8*), the expression for population reaction times can be obtained as

$$T_{th} \gg A(w)\frac{1}{aN}B_0\left(\frac{1}{w}\right)^{-(D-1)}\exp\left(\frac{D}{w}\right)\frac{\overline{\lambda^{-3}}^2}{\overline{\lambda^{-2}}^3}, \tag{46}$$

where $A(w)$ is a function of $w$. Lastly, by casting *Equation 46* in terms of the scale factor $c$, and fitting using (for example) least square regression, we obtain

$$T_{th} \approx K_1 A(w)\frac{1}{aM}\frac{\exp(D/w)}{B_0(1/w)^{(D-1)}}\frac{\left(\sum\limits_{j=0}^{L-1}c^{-3j}\right)^2}{\left(\sum\limits_{j=0}^{L-1}c^{-2j}\right)^3} + K_2, \tag{47}$$

where $M$ is the number of neurons per module, and $K_1$ and $K_2$ are constants. Note that in the simulations, $w$ is fixed and $A(w)$ can therefore be incorporated into $K_1$.

## Acknowledgements

We thank the reviewers and editors for their helpful comments on improving the manuscript and Dr. Pascal Helson for proofreading the manuscript.

## Additional information

### Funding

| Funder | Grant reference number | Author |
|---|---|---|
| Digital Futures | | Movitz Lenninger<br>Mikael Skoglund<br>Pawel Andrzej Herman<br>Arvind Kumar |
| Vetenskapsrådet | | Arvind Kumar |
| Institute of Advanced Studies | Fellowship | Arvind Kumar |

The funders had no role in study design, data collection and interpretation, or the decision to submit the work for publication.

### Author contributions

Movitz Lenninger, Conceptualization, Data curation, Software, Formal analysis, Validation, Investigation, Visualization, Methodology, Writing - original draft, Writing – review and editing; Mikael Skoglund, Supervision, Funding acquisition, Writing – review and editing; Pawel Andrzej Herman, Resources, Supervision, Funding acquisition, Writing – review and editing; Arvind Kumar, Conceptualization, Supervision, Methodology, Writing - original draft, Writing – review and editing

### Author ORCIDs

Movitz Lenninger http://orcid.org/0000-0002-6165-4900
Pawel Andrzej Herman http://orcid.org/0000-0001-6553-823X
Arvind Kumar https://orcid.org/0000-0002-8044-9195

### Decision letter and Author response

Decision letter https://doi.org/10.7554/eLife.84531.sa1
Author response https://doi.org/10.7554/eLife.84531.sa2

## Additional files

**Supplementary files**
• MDAR checklist

**Data availability**
Code has been made publicly available on Github (https://github.com/movitzle/Short_Decoding_Time, copy archived at *Lenninger, 2023*).

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

## Appendix 1

### Tuning curves and spike count model

In the paper, we study the representation of a multidimensional stimulus $s = (s_1, ..., s_D)$. For simplicity, it is assumed that the range of the stimulus in each dimension is equal, such that $s_j \in [0, R)$ for all $j \in \{1, ..., D\}$ for some stimulus range $R$ (in the main text, we consider $R = 1$ for simplicity). Note that these assumption does not qualitatively change the results. Furthermore, we assume that the tuning curves were circular (von Mises) tuning curves

$$f_i(s) = a_i \prod_{j=1}^{D} \exp\left( \frac{1}{w}\left( \cos(\frac{2\pi}{\lambda_i R}(s_j - s'_{i,j})) - 1 \right) \right) + b = a_i \prod_{j=1}^{D} q_{i,j}(s) + b, \tag{A1.1}$$

where $a_i$ is the peak amplitude of the stimulus-related tuning curve of neuron $i$, $w$ is a width scaling parameter, $\lambda_i$ defines the spatial period of the tuning, $s'_{j,i}$ determines the location of the firing field(s) in the $j$:th dimension, and $b$ determines the amount of background activity. The amplitude parameters $a_i$ were tuned such that all tuning curves had the same firing rate when averaged across all stimulus conditions (see Supplementary *Equation A1.5*).

It is possible to reparametrize the stimuli into a phase variable, $\phi = \frac{s_j}{R}$. In the article, calculations and numerical simulation are based on phase variables $\phi$. This only changes the MSE and Fisher information by a constant scaling $\frac{1}{R^2}$. As we are interested in comparing the minimal decoding times, not the absolute values of the MSE, we can drop the "unnormalized" stimulus $s$. The tuning curves in Supplementary *Equation A1.1* can thus be rewritten using the phase variable $\phi$ as

$$f_i(\phi) = a_i \prod_{j=1}^{D} \exp\left( \frac{1}{w}\left( \cos(\frac{2\pi}{\lambda_i}(\phi_j - \phi'_{i,j})) - 1 \right) \right) + b = a_i \prod_{j=1}^{D} q_{i,j}(\phi) + b. \tag{A1.2}$$

Given stimulus condition $\mathbf{s}$ (or $\phi$) and decoding time $T$, the spike count of each neuron was independently sampled from a Poisson distribution with rate $Tf_i(\mathbf{s})$. Thus, the probability of observing a particular spike count pattern $\mathbf{r} = (r_1, ..., r_N)$ given $\mathbf{s}$ is

$$p(\mathbf{r}|\mathbf{s}) = \prod_{i=1}^{N} p(r_i|\mathbf{s}) = \prod_{i=1}^{N} \frac{(Tf_i(\mathbf{s}))^{r_i} \exp(-Tf_i(\mathbf{s}))}{r_i!}. \tag{A1.3}$$

### Adjusting amplitudes

In order to make a fair comparison of decoding times across populations, we constrain each neuron to have the same average firing rate across the stimulus domain, $\bar{f}$. The average firing rate over the stimulus domain is

$$\bar{f}_i = b + a_i \frac{1}{R^D} \int_0^R ... \int_0^R \prod_{j=1}^{D} q_{i,j}(\phi_j) d\phi_1 ... d\phi_D. \tag{A1.4}$$

Thus, given a desired stimulus-evoked firing rate, $\overline{f_{stim}}$, over a normalized stimulus range ($R = 1$), the amplitudes will be set to

$$a_i = \frac{\overline{f_{stim}}}{\int_0^1 ... \int_0^1 \prod_{j=1}^{D} q_{i,j}(\phi_j) d\phi_1 ... d\phi_D}. \tag{A1.5}$$

Note that the integrals in *Equation A1.5* are analytically solvable whenever the relative spatial frequency $\xi_i = 1/\lambda_i$ is a positive integer, in which case we have

$$\int_0^1 ... \int_0^1 \prod_{j=1}^{D} \exp\left( \frac{1}{w}\left( \cos(\frac{2\pi}{\lambda_i}(\phi_j - \phi'_{i,j})) - 1 \right) \right) d\phi_1 ... d\phi_D = B_0\left(\frac{1}{w}\right)^D \exp\left(-\frac{D}{w}\right) \tag{A1.6}$$

regardless of $\phi'_{i,j}$, here $B_0(\cdot)$ is the modified Bessel function of the first kind. In simulations, $\overline{f_{stim,i}} = \overline{f_{stim}}$ was set such that tuning curves with integer spatial frequencies ($1/\lambda$) have amplitudes of 20 sp/s, that is,

$$\overline{f_{stim}(D)} = 20B_0 \left( \frac{1}{w} \right)^D \exp\left( -\frac{D}{w} \right).$$

## Appendix 2

### Fisher information and the Cramér-Rao bound

Assuming a one-dimensional variable, the Cramér-Rao bound gives a lower bound on the MSE of any estimator $G$

$$\mathbb{E}[(G(\mathbf{r}) - s)^2] \geq \frac{[1 + b'_G(s)]^2}{J(s)} + b_G(s)^2, \tag{A2.1}$$

where $b_G(s) = \mathbb{E}[G(\mathbf{r}) - s]$ is the bias of the estimator $G$ and $J(s)$ is Fisher information, defined as

$$J(s) = \mathbb{E}\left[\frac{\partial}{\partial s} \log p(\mathbf{r}|s)\right]^2 = -\mathbb{E}\left[\frac{\partial^2}{\partial s^2} \log p(\mathbf{r}|s)\right] \tag{A2.2}$$

where the last equality holds if $p(\mathbf{r}|s)$ is twice differentiable and the neural responses are conditionally independent (*Lehmann and Casella, 1998*). Assuming an unbiased estimator, the bound can be simplified to

$$\mathbb{E}[(G(s) - s)^2] = \mathrm{Var}(G(\mathbf{r})) \geq \frac{1}{J(s)}. \tag{A2.3}$$

For multi-parameter estimation, let $J(\mathbf{s})$ denote the Fisher information matrix, with elements defined analogously to Supplementary *Equation A2.2*

$$J_{k,l}(\mathbf{s}) = -\mathbb{E}\left[\frac{\partial^2}{\partial s_k \partial s_l} \log p(\mathbf{r}|\mathbf{s})\right], \tag{A2.4}$$

then (for unbiased estimators) the Cramér-Rao bound is instead stated as the following matrix inequality (*Lehmann and Casella, 1998*)

$$\mathrm{Cov}(G) = \Sigma \geq J^{-1}(\mathbf{s}) \tag{A2.5}$$

in the sense that the difference $\Sigma - J^{-1}(\mathbf{s})$ is a positive semi-definite matrix. Thus, this implies the following lower bound for MSE of the k:th term

$$\mathrm{Var}(G^{(s_k)}) = \Sigma_{k,k} \geq \{J^{-1}(\mathbf{s})\}_{k,k} \tag{A2.6}$$

where $G^{(s_k)} = \hat{s}_k$, that is, the estimation of $s_k$ using estimator $G$. Note that if $J(\mathbf{s})$ is a diagonal matrix, that is, $\{J(\mathbf{s})\}_{j,k} = 0$ for all $j \neq k$, then the following also holds

$$\{J^{-1}(\mathbf{s})\}_{k,k} = \{J_{k,k}(\mathbf{s})\}^{-1}. \tag{A2.7}$$

For the tuning curves defined in Supplementary *Equation A1.1*, the diagonal elements of the Fisher information matrix can be analytically solved assuming $a_i \sim a$ within each module (see Supplementary *Equation A2.19*)

$$J_{k,k}(s) \approx (2\pi)^2 TN \frac{a}{R^2 w} B_0 \left(\frac{1}{w}\right)^{D-1} \exp\left(-\frac{D}{w}\right) B_1 \left(\frac{1}{w}\right) \overline{\lambda^{-2}} \tag{A2.8}$$

where the bar indicates the sample average across modules and $R$ is the stimulus range (note that in the main text, we assume $R = 1$). The off-diagonal elements, on the other hand, can be shown to be 0 (see below). Thus we have equality in the last inequality of Supplementary *Equation A2.6*, and the MSE for each stimulus dimension is lower bounded by the inverse of Supplementary *Equation A2.8*.

### Approximating Fisher information

To analytically approximate the Fisher information for a given neural population, we will neglect the impact of ongoing activity $b$. Then, the tuning curves in Supplementary *Equation A1.1* factorize as $f_i(s) = a q_{1,i}(s_1)...q_{D,i}(s_D)$ and the log-likelihood for $N$ neurons with conditionally independent spike counts becomes

$$\log p(\mathbf{r}|\mathbf{s}) = \sum_{i=1}^{N} r_i \log(Tf_i(\mathbf{s})) - Tf_i(\mathbf{s}) - \log r_i! \tag{A2.9}$$

By taking the second derivatives w.r.t. stimulus dimension, we get for $k = l$:

$$\frac{\partial^2}{\partial s_k^2} \log p(\mathbf{r}|\mathbf{s}) = \dots = \sum_{i}^{N} r\left(\frac{q''_{k,i}}{q_{k,i}} - \left(\frac{q'_{k,i}}{q_{k,i}}\right)^2\right) - Tf_i \frac{q''_{k,i}}{q_{k,i}} \tag{A2.10}$$

and for $k \neq l$

$$\frac{\partial^2}{\partial s_k \partial s_l} \log p(\mathbf{r}|\mathbf{s}) = \dots = \sum_{i}^{N} -Tf_i \frac{q'_{k,i} q'_{l,i}}{q_{k,i} q_{l,i}}. \tag{A2.11}$$

Consequently, the elements of the Fisher information matrix are given by

$$J_{k,k}(\mathbf{s}) = -\mathbb{E}\left[\frac{\partial^2}{\partial s_k^2} \log p(\mathbf{r}|\mathbf{s})\right] = \sum_{i=1}^{N} Tf_i(\mathbf{s}) \left(\frac{q'_{k,i}(s_k)}{q_{k,i}(s_k)}\right)^2 \tag{A2.12}$$

and for $k \neq l$

$$J_{k,l}(\mathbf{s}) = -\mathbb{E}\left[\frac{\partial^2}{\partial s_l \partial s_k} \log p(r|\mathbf{s})\right] = \sum_{i=1}^{N} Tf_i(\mathbf{s}) \frac{q'_{k,i}(s_k) q'_{l,i}(s_l)}{q_{k,i}(s_k) q_{l,i}(s_l)}. \tag{A2.13}$$

To simplify calculations, it is possible to reparametrize the stimulus as in Supplementary *Equation A1.2* using the formula for Fisher information under reparametrization (*Lehmann and Casella, 1998*)

$$J_{k,l}(\phi) = \sum_{m} \sum_{n} \frac{\partial s_m}{\partial \phi_k} \frac{\partial s_n}{\partial \phi_l} J_{k,l}(\mathbf{s}) = R^2 J_{k,l}(\mathbf{s}) \tag{A2.14}$$

to obtain

$$J(\mathbf{s}) = \frac{1}{R^2} J_{k,l}(\phi). \tag{A2.15}$$

We can approximate the elements of the Fisher information matrix $J(\phi)$ in the limit of large $N$ by replacing the sums with integrals, for example,

$$J_{k,k}(\phi) = \sum_{i=1}^{N} Tf_i(\phi) \left(\frac{q'_{k,i}(\phi_k)}{q_{k,i}(\phi_k)}\right)^2 \approx$$

$$\approx T \sum_{j=1}^{L} \frac{L}{\lambda_j^D} a_j \int_{\phi_1 - \frac{1}{2}\lambda_j}^{\phi_1 + \frac{1}{2}\lambda_j} \cdots \int_{\phi_D - \frac{1}{2}\lambda_j}^{\phi_D + \frac{1}{2}\lambda_j} \left[\prod_{p=1}^{D} \exp\left(\frac{1}{w}\left(\cos(\frac{2\pi}{\lambda_j}(\phi_p - \phi'_p)) - 1\right)\right)\right] \frac{(2\pi)^2 \sin^2(\frac{2\pi}{\lambda_j}(\phi_k - \phi'_k))}{\lambda_j^2 w^2} \tag{A2.16}$$

where $L$ is the number of distinct modules, $M$ is the number of neurons in each module, $d\phi' = d\phi'_1 \dots d\phi'_D$, and the D-dimensional integral is taken over the interval $[\phi_p - \frac{1}{2}\lambda_j, \phi_p + \frac{1}{2}\lambda_j)$ along each dimension. Making the variable substitution $\theta_p = \frac{2\pi}{\lambda_j}(\phi_p - \phi'_p)$ for $p = \{1, \dots, D\}$ we have

$$J_{k,k}(\phi) \approx MTa \sum_{j=1}^{L} \frac{1}{\lambda_j^D} \int_{\pi}^{-\pi} \cdots \int_{\pi}^{-\pi} \left[\prod_{p=1}^{D} \exp\left(\frac{1}{w}\left(\cos(\theta_p) - 1\right)\right)\right] \frac{(2\pi)^2 \sin^2(\theta_k)}{\lambda_j^2 w^2} \frac{\lambda_j^D}{(-1)^D (2\pi)^D} d\theta =$$

$$= \dots = \frac{(2\pi)^2 NTa}{w} B_0 \left(\frac{1}{w}\right)^{D-1} B_1 \left(\frac{1}{w}\right) \exp\left(-\frac{D}{w}\right) \overline{\lambda^{-2}} \tag{A2.17}$$

where the sample average is taken over the population's distribution of spatial frequencies and $B_\alpha(\cdot)$ is the modified Bessel function of the first kind. Similar calculations for the case $k \neq l$ yield

$$J_{k,l}(\phi) \quad = \dots \approx MT \sum_{j=1}^{L} \frac{a}{w^2} \exp\left(-\frac{D}{w}\right) B_0\left(\frac{1}{w}\right)^{D-2} \int_{-\pi}^{\pi} \frac{1}{\lambda_j} \sin(\theta_k) \exp\left(\frac{1}{w}\cos(\theta_k)\right) d\theta_k$$

$$\int_{-\pi}^{\pi} \frac{1}{\lambda_j} \sin(\theta_l) \exp\left(\frac{1}{w}\cos(\theta_l)\right) d\theta_l = 0.$$

(A2.18)

Thus, the stimulus parameters will be asymptotically orthogonal for all of the populations considered in this paper. That is, the covariance matrix will be diagonal. The per-neuron average contribution to each diagonal element of the Fisher information matrix, as reported in the main text, is, therefore

$$\bar{J}_{k,k}(s) \approx T \frac{(2\pi)^2 a}{R^2 w} B_0\left(\frac{1}{w}\right)^{D-1} B_1\left(\frac{1}{w}\right) \exp\left(-\frac{D}{w}\right) \overline{\lambda^{-2}}.$$

(A2.19)

## Appendix 3

## Maximum of the joint likelihood function (2 module case)

Assuming that the responses of the two modules are independent, the joint likelihood function $p(\mathbf{r}|s)$ can be decomposed into a product of the likelihood functions of the two modules. Using the approximation of each $Q_1(\mathbf{r}|s)$ and $Q_2(\mathbf{r}|s)$ as Gaussian sums (see Materials and methods), we have the following

$$
\begin{aligned}
p(\mathbf{r}|s) \quad &= Q_1(s)Q_2(s) \approx \hat{Q}_1(s)\hat{Q}_2(s) \\
&= A_1 \sum_{n_1=-K_1}^{K_1} \exp\left(-\frac{J_1}{2}\left(s - (s_{ML}^{(1)} + n_1\lambda_1 R)\right)^2\right) A_2 \sum_{n_2=-K_2}^{K_2} \exp\left(-\frac{J_2}{2}\left(s - (s_{ML}^{(2)} + n_2\lambda_2 R)\right)^2\right)
\end{aligned}
$$

(A3.1)

Thus, the contribution of the $p$:th and $q$:th mode of $\hat{Q}_1$ and $\hat{Q}_2$ to the joint likelihood function is

$$
\begin{aligned}
\hat{Q}_1^p \hat{Q}_2^q \quad &= A_1 A_2 \exp\left(-\frac{\Sigma_1}{2}\left(s - (s_{ML}^{(1)} + p\lambda_1 R)\right)^2\right) \exp\left(-\frac{\Sigma_2}{2}\left(s - (s_{ML}^{(2)} + q\lambda_2 R)\right)^2\right) \\
&= A_1 A_2 \exp\left(-\frac{\Sigma_1}{2}(s - s_p)^2 - \frac{\Sigma_2}{2}(s - s_q)^2\right)
\end{aligned}
$$

(A3.2)

where we in the last step renamed $s_{ML}^{(1)} + p\lambda_1 R$ and $s_{ML}^{(2)} + q\lambda_2 R$ to $s_p$ and $s_q$, respectively. Unless the width $w$ of the tuning curves or the range $R$ is very large, all the modes of $\hat{Q}_1$ and $\hat{Q}_2$, respectively, are well separated (see the end of the section). Thus, it is a reasonable approximation that the maximum of $p(\mathbf{r}|s) = Q_1(s)Q_2(s)$ is defined by the maximum of $\hat{Q}_1^p(s)\hat{Q}_2^q(s)$ across all combinations of $p$ and $q$. Each combination $\hat{Q}_1^p(s)\hat{Q}_2^q(s)$ reaches its maximum for some stimulus $s_{p,q}^*$:

$$
s_{p,q}^* = \arg\max_s \hat{Q}_1^p(s)\hat{Q}_2^q(s) = \arg\min_s \frac{\Sigma_1}{2}(s - s_p)^2 + \frac{\Sigma_2}{2}(s - s_q)^2
$$

(A3.3)

Taking the derivative w.r.t. $s$ on the rightmost terms and solving gives

$$
s_{p,q}^* = \frac{\Sigma_1 s_p + \Sigma_2 s_q}{\Sigma_1 + \Sigma_2}.
$$

(A3.4)

Thus, using $\delta_{p,q} = (s_p - s_q)$, the maximal value of each pair $\hat{Q}_1^p(s)\hat{Q}_2^q(s)$ is

$$
\begin{aligned}
\hat{Q}_1^p(s_{p,q}^*)\hat{Q}_2^q(s_{p,q}^*) \quad &= A_1 A_2 \exp\left(-\frac{J_1}{2}\left(s_{p,q}^* - s_p\right)^2 - \frac{J_2}{2}\left(s_{p,q}^* - s_q\right)^2\right) = \dots = \\
&= A_1 A_2 \exp\left(-\frac{J_1}{2}\left(\frac{-J_2\delta_{p,q}}{J_1 + J_2}\right)^2 - \frac{J_2}{2}\left(\frac{J_1\delta_{p,q}}{J_1 + J_2}\right)^2\right) = \dots = \\
&= A_1 A_2 \exp\left(-\frac{1}{2}\frac{J_1 J_2}{J_1 + J_2}\delta_{p,q}^2\right)
\end{aligned}
$$

(A3.5)

Thus, the maximum likelihood choice will approximately be $s_{p,q}^*$ for the $p$:th and $q$:th mode with the smallest $\delta_{p,q}^2$, that is, the smallest distance between the modes. Lastly, all modes of $\hat{Q}_1$ and $\hat{Q}_2$, respectively, need to be sufficiently separated such that no two pairs of $p$ and $q$ reinforce each other. However, it is well known the full width at half maximum for a Gaussian function is $\mathrm{FWHM} = 2\sqrt{2\ln 2}\sigma_i$, where for our functions $\hat{Q}_1^p$ and $\hat{Q}_2^q$, $\sigma_1 = 1/\sqrt{J_1}$ and $\sigma_2 = 1/\sqrt{J_2}$. Thus, given the expression for Fisher information in *Equation A2.8*, the FWHM can be expressed as

$$
FWHM = \frac{2}{\pi}\lambda_i \sqrt{\frac{\ln(2)wR^2}{2aTMB_0(1/w)^{D-1}B_1(1/w)\exp(-D/w)}}
$$

(A3.6)

Thus, for the modes to be separated, it is reasonable to require that the FWHM is no longer than one period length of the module, that is, $\lambda_i$. Hence, we have that

$$
\frac{2}{\pi}\lambda_i \sqrt{\frac{\ln(2)wR^2}{2aTMB_0(1/w)^{D-1}B_1(1/w)\exp(-D/w)}} < \lambda_i
$$

(A3.7)

Rewriting this into a bound on the time $T$ needed for the assumption of separation, we get

$$T > \left(\frac{\pi}{2}\right)^2 \frac{\ln(2)wR^2}{2aMB_0(1/w)^{D-1}B_1(1/w)\exp(-D/w)} \tag{A3.8}$$

For the parameters used in our simulations, this is satisfied very fast, on the order of tens of microseconds. However, note that the assumption of each module providing efficient estimates, which is a prerequisite for these approximations, requires significantly longer time scales. Thus, the assumption that the individual modes of $\hat{Q}_1$ and $\hat{Q}_2$ are well-separated is, in our case, not likely to be a restrictive assumption.

## Appendix 4

### Calculate $M_{k,l,m}$

To approximate the minimal decoding time, we need to calculate (see *Equations 38; 39*, main text)

$$M_{k,l,m}(\mathbf{s}) = \frac{\partial^3}{\partial s_m \partial s_l \partial s_k} \log p(\mathbf{r}|\mathbf{s}) \approx \mathbb{E}\left[\frac{\partial^3}{\partial s_m \partial s_l \partial s_k} \log p(\mathbf{r}|\mathbf{s})\right]. \tag{A4.1}$$

For $k \neq l \neq m$, using Supplementary *Equation A1.1*, *Equation A1.2*, *Equation A1.3*, we have

$$\frac{\partial^3}{\partial s_m \partial s_l \partial s_k} \log p(\mathbf{r}|\mathbf{s}) = -\sum_{i=1}^{N} Tf_i(\mathbf{s}) \frac{q'_{k,i}(s_k) q'_{l,i}(s_l) q'_{m,i}(s_m)}{q_{k,i}(s_k) q_{l,i}(s_l) q_{m,i}(s_m)} \tag{A4.2}$$

Thus, $M_{k,l,m}$ for $k \neq l \neq m$ becomes

$$\begin{aligned}
M_{k,l,m}(\mathbf{s}) \quad &= -\sum_{i=1}^{N} Tf_i(\mathbf{s}) \frac{q'_{k,i}(s_k) q'_{l,i}(s_l) q'_{m,i}(s_m)}{q_{k,i}(s_k) q_{l,i}(s_l) q_{m,i}(s_m)} \approx \\
&\approx \sum_{j=1}^{L} \frac{M}{(\lambda_j R)^D} \int_{s_1 - \frac{R}{2}\lambda_j}^{s_1 + \frac{R}{2}\lambda_j} \cdots \int_{s_D - \frac{R}{2}\lambda_j}^{s_D + \frac{R}{2}\lambda_j} Ta\left[\prod_{j=1}^{D} \exp\left(\frac{1}{w}\left(\cos\left(\frac{2\pi}{\lambda_j R}(s_j - s'_j)\right) - 1\right)\right)\right] \\
&\quad \frac{(2\pi)^3}{\lambda_j^3 w^3} \sin\left(\frac{2\pi}{\lambda_j R}(s_k - s'_k)\right) \sin\left(\frac{2\pi}{\lambda_j R}(s_l - s'_l)\right) \sin\left(\frac{2\pi}{\lambda_j R}(s_m - s'_m)\right) d\mathbf{s}' = 0
\end{aligned} \tag{A4.3}$$

as odd functions over even intervals integrate to zero. For $k \neq l = m$ (note that $k = l \neq m$ and $k = m \neq l$ follows by symmetry) we have

$$\frac{\partial^3}{\partial s_l^2 \partial s_k} \log p(\mathbf{r}|\mathbf{s}) = -\sum_{i=1}^{N} Tf_i(\mathbf{s}) \frac{q'_{k,i}(s_k) q''_{l,i}(s_l)}{q_{k,i}(s_k) q_{l,i}(s_l)} \tag{A4.4}$$

and hence,

$$\begin{aligned}
M_{k,l,l} \quad &= -\sum_{i=1}^{N} Tf_i(\mathbf{s}) \frac{q'_{k,i}(s_k) q''_{l,i}(s_l)}{q_{k,i}(s_k) q_{l,i}(s_l)} \approx \\
&\approx -\sum_{j=1}^{L} \frac{M}{(\lambda_j R)^D} \int_{s_1 - \frac{R}{2}\lambda_j}^{s_1 + \frac{R}{2}\lambda_j} \cdots \int_{s_D - \frac{R}{2}\lambda_j}^{s_D + \frac{R}{2}\lambda_j} Ta\left[\prod_{j=1}^{D} \exp\left(\frac{1}{w}\left(\cos\left(\frac{2\pi}{\lambda_j R}(s_j - s'_j)\right) - 1\right)\right)\right] \\
&\quad \frac{(2\pi)^3}{\lambda_j^3 w^3} \sin\left(\frac{2\pi}{\lambda_j R}(s_k - s'_k)\right)\left(w\cos\left(\frac{2\pi}{\lambda_j R}(s_l - s'_l)\right) - \sin^2\left(\frac{2\pi}{\lambda_j R}(s_l - s'_l)\right)\right) d\mathbf{s}' = 0.
\end{aligned} \tag{A4.5}$$

Lastly, for $k = l = m$ we have,

$$\frac{d^3}{ds_k^3} \log p(\mathbf{r}|\mathbf{s}) = \sum_{i=1}^{N} (r_i - Tf_i(\mathbf{s})) \frac{q'''_{k,i}(s_k)}{q_{k,i}(s_k)} - 3r_i \frac{q'_{k,i}(s_k) q''_{k,i}(s_k)}{q_{k,i}(s_k)^2} + 2r_i \left(\frac{q'_{k,i}(s_k)}{q_{k,i}(s_k)}\right)^3. \tag{A4.6}$$

Thus $M_{k,k,k}(\mathbf{s}^*)$, becomes

$$\begin{aligned}
M_{k,k,k}(\mathbf{s}^*) \quad &= \mathbb{E}_{\mathbf{r}\sim\text{Poiss}(f(\mathbf{s}^\circ))}\left[\frac{d^3}{ds_k^3} \log p(\mathbf{r}|\mathbf{s})\Big|_{\mathbf{s}=\mathbf{s}^*}\right] \\
&= \sum_{i=1}^{N} (Tf_i(\mathbf{s}^\circ) - Tf_i(\mathbf{s}^*)) \frac{q'''_{k,i}(s_k^*)}{q_{k,i}(s_k^*)} - 3Tf_i(\mathbf{s}^\circ) \frac{q'_{k,i}(s_k^*) q''_{k,i}(s_k)}{q_{k,i}(s_k)^2} + 2Tf_i(\mathbf{s}^\circ)\left(\frac{q'_{k,i}(s_k)}{q_{k,i}(s_k)}\right)^3.
\end{aligned} \tag{A4.7}$$

Each term above have a dependence on $\sin(\frac{2\pi}{\lambda_i R}(s_k^* - s'_{k,i}))$, with an odd power. Therefore, when multiplying with $f(\mathbf{s}^*)$ and integrating as above, these terms vanish. Hence, we can focus only on the terms including $f(\mathbf{s}^\circ)$. After some calculus manipulation, it is possible to reduce the expression to include only $T\sin(\frac{2\pi}{\lambda_i R}(s_k^* - s'_{k,i}))f_i(\mathbf{s}^\circ)$ (for all $i$).

$$M_{k,k,k}(\mathbf{s}^*) \approx T \sum_{j=1}^{L} a \frac{M}{(\lambda_j R)^D} \frac{(2\pi)^3}{\lambda_j^3 R^3 w} \int_{s_1^* - \frac{R}{2}\lambda_j}^{s_1^* + \frac{R}{2}\lambda_j} \cdots \int_{s_D^* - \frac{R}{2}\lambda_j}^{s_D^* + \frac{R}{2}\lambda_j} \sin\left( \frac{2\pi(s_k^* - s_k')}{\lambda_j R} \right) \times$$

$$\times \prod_{j=1}^{D} \exp\left( \frac{1}{w}\left( \cos\left( \frac{2\pi}{\lambda_i R}(s_j^\circ - s_j') \right) - 1 \right) \right) d\mathbf{s}' =$$

$$= T \sum_{j=1}^{L} a \frac{M}{(\lambda_j R)} \frac{(2\pi)^3}{\lambda_j^3 R^3 w} \exp\left( -\frac{D}{w} \right) B_0\left( \frac{1}{w} \right)^{(D-1)} \times \tag{A4.8}$$

$$\times \int_{s_k^* - \frac{R}{2}\lambda_j}^{s_k^* + \frac{R}{2}\lambda_j} \sin\left( \frac{2\pi(s_k^* - s_k')}{\lambda_j R} \right) \exp\left( \frac{1}{w}\left( \cos\left( \frac{2\pi}{\lambda_i R}(s_k^\circ - s_k') \right) - 1 \right) \right) ds_k'$$

Unfortunately, as this integral includes both $s_k^*$ and $s_k^\circ$, no simple expression can be obtained. Using the variable substitution $\theta_k^* = \frac{2\pi}{\lambda_j R}(s_k^* - s_k')$, we can simplify it slightly to

$$M_{k,k,k}(\mathbf{s}^*) \approx TaM \frac{(2\pi)^2}{R^3 w} \exp\left( -\frac{D}{w} \right) B_0\left( \frac{1}{w} \right)^{(D-1)} \sum_{j=1}^{L} \frac{1}{\lambda_j^3} \times$$

$$\times \int_{-\pi}^{\pi} \sin(\theta_k^*) \exp\left( \frac{1}{w}\left( \cos\left( \theta_k^* + \frac{2\pi}{\lambda_i R}(s_k^\circ - s_k^*) \right) - 1 \right) \right) d\theta_k^* \tag{A4.9}$$

Instead, we focus on the difference $\phi_j^* = \phi_k(\lambda_j) = s_k^\circ - s_k^*$, which maximizes the above integral for each module. Thus, all integrals can be upper bounded by a constant $C^*$, yielding the upper bound

$$M_{k,k,k}(\mathbf{s}^*) \leq M^* = (2\pi)^2 C^* \frac{TaN}{R^3 w} \exp\left( -\frac{D}{w} \right) B_0\left( \frac{1}{w} \right)^{(D-1)} \overline{\lambda^{-3}} \tag{A4.10}$$

Note that the constant $C^*$ can be incorporated into the regression coefficient $K_1$ in **Equation 47** and that the stimulus range, $R$, is assumed to be $R = 1$ in the main text.

## Appendix 5

### Approximating minimal required spike count

Given the approximation of minimal decoding time in *Equation 9* (main text), we seek to reformulate the approximation in terms of the required total spike count, instead. The average total spike count for a given population and stimulus condition is

$$\mu(\mathbf{s}) = \mathbb{E}_r \left[ \sum_{i=1}^{N} r_i \,\middle|\, \mathbf{s} \right] = \sum_{i=1}^{N} T f_i(\mathbf{s}) \tag{A5.1}$$

where $T$ is the decoding time. Thus, the average spike count over both stimulus conditions (assuming uniformly distributed stimulus and integer spatial frequencies) and trials for the entire population is

$$\mu = \mathbb{E}_\mathbf{s} \left[ \mathbb{E}_\mathbf{r} \left[ \sum_{i=1}^{N} r_i \,\middle|\, \mathbf{s} \right] \right] = \frac{1}{R^D} \int_0^R \cdots \int_0^R \sum_{i=1}^{N} T f_i(\mathbf{s}) d\mathbf{s} = NT(aB_0(1/w)^D \exp(-D/w) + b). \tag{A5.2}$$

Consequently, the number of spikes evoked by the stimulus-related tuning of the population is

$$\mu_{stim} = NTaB_0(1/w)^D \exp(-D/w). \tag{A5.3}$$

Inserting Supplementary *Equation A5.3* into *Equation 46* (main text) reveals the number of stimulus-evoked spikes, $\mu_{stim}^*$, the population must produce before reaching the predicted lower bound

$$\mu_{stim}^* \gg A(w) B_0(1/w) \frac{\overline{\lambda^{-3}}^2}{\overline{\lambda^{-2}}^3}. \tag{A5.4}$$

