## [Editor Report]

This fundamental study provides important insight into coding strategies in sensory areas. The study was well done, and the analysis and simulations were highly convincing. This study should be of particular interest to anybody who cares about efficient coding.

---

## [Decision Letter]

**Decision letter after peer review:**

Thank you for submitting your article "Are single-peaked tuning curves tuned for speed rather than accuracy?" for consideration by *eLife*. Your article has been reviewed by 2 peer reviewers, one of whom is a member of our Board of Reviewing Editors, and the evaluation has been overseen by Panayiota Poirazi as the Senior Editor. The following individual involved in review of your submission has agreed to reveal their identity: Stefano Panzeri (Reviewer #2).

Essential revisions:

While this paper makes an extremely important point, one that should definitely be communicated to the broader community, in our opinion it suffers from two (relatively easily fixable) problems:

I. Unless one is an expert, it's very hard to extract why multi-peaked tuning curves lead to catastrophic errors.

II. It's difficult to figure out under what circumstances multi-peaked tuning curves are bad. This is important, because there are a lot of neurons in sensory cortex, and one would like to know whether periodic tuning curves are really a bad idea there.

And here are the fixes:

I. Figure 1c seems like a missed opportunity to explain what's really going on, which is that on any particular trial the positions of the peaks of the log likelihood can shift in both phase and amplitude (with phase being more important). The reason it's a missed opportunity is that Figure 1c shows the average log likelihood, which makes it hard to understands why there are catastrophic errors. It would really help if Figure 1c were expanded into its own large figure, with sample log likelihoods showing catastrophic errors for multi-peaked tuning curves but not for single-peaked ones. You could also indicate why, when multi-peaked tuning curves do give the right answer, the error tends to be small.

II. What the reader really wants to know is: would sensory processing in real brains be more efficient if multi-peaked tuning curves were used? That's certainly hard to answer in all generality, but you could make a comparison between a code with single-peaked tuning curves and a _good_ code with multi-peaked tuning curves. Our guess is that a good code would have λ_1=1 and c around 0.5 (but not, of course, exactly 1/2; you could use the module ratio the grid cell people came up with -- we think 1/sqrt(2) -- although we doubt if it matters much). What would be great would be a relatively simple relation, possibly fitted empirically to simulations, that told us when multi-peaked tuning curves are better than single-peaked one, given the number of neurons, background and peak firing rates, and dimensionality. Of course, this may be impossible, but if so it's important to state that up front, so that people know how difficult this is.

In addition, we have a number of comments, all designed to improve the paper. Feel free to disagree with any of them; we don't feel massively strongly about them. Also, if you implement them, no reason to provide a reply -- we're sensitive to the fact that replies to the reviewer are becoming longer than the paper!

*Reviewer #1 (Recommendations for the authors):*

1. When catastrophic errors are possible, we agree that it's very hard to find a good measure of error. RMS error clearly isn't very good. However, it's not clear minimal decoding time is any better, since it'd defined in terms of RMS error. Finding a good measure seems critical when comparing codes, given that Fisher info can change by 5 orders of magnitude while decoding time changes by only 1 (e.g., Figure 4d). Moreover, we could be wrong, but our intuition is that when λ_1=1, catastrophic errors aren't much of an issue. Is that correct? If so, time to threshold (defined to be time for the RMS error to be twice that predicted by the inverse of the Fisher info) could be a bad measure.

Here we don't have any really great suggestions. But you might try something like [<(error)^k>/((2k-1)!!)]^{1/k} for k larger than 2 and even (the factor of (2k-1)!!) makes this equal to the RMS error for Gaussian distributions. But other measures are possible.

2. Are you using periodic boundary conditions? Tuning curves are clearly periodic, but decoding near the boundaries is a bit tricky. For instance, the ML estimator can put the decoded value outside the range [0, 1). This should be clarified.

3. It should be clear that the expression for Fisher info (Equation 3) applies when a_i=a and b=0.

4. Figure 1a: it would be helpful to indicate the location of s_1 and s_2 in the top panel.

Also, isn't Figure 1a a bit misleading? The problem isn't a small number of spikes; it's a phase shift in the log likelihood.

5. l 99-100: "For the largest mode of the joint likelihood function to also be centered close to the true stimulus condition, the distance δ between s^(1)_ML and s^(2)_ML must be smaller than between any other pair of modes of Q_1 and Q_2." We can see why this might be reasonable, but it's not totally obvious that the distance between the modes is the only thing that matters. This needs to be shown.

6. l 100-101: "Thus, to avoid catastrophic errors, δ must be smaller than some largest allowed distance δ* (see Methods for calculation of δ*)." This makes it seems like δ* is something complicated. But in fact it's exactly half the smallest distance between any other pair of modes. Why not say so?

7. l 120-1: "estimates can cause catastrophic shifts in the joint likelihood function. For single-peaked tuning curves (λ_1 = 1), however, only small scale factors c can pose such problems." Should "For single-peaked tuning curves" be "When the first mode has a single peak". If not, we're lost.

8. l 128: "Furthermore, only populations with unambiguous codes over the stimulus interval were included [22]." Can't parse.

9. Equation 8 seems unlikely to be correct, given that it doesn't depend on c, which we know is critical (as shown in Figure 1e). However, if you do keep it, it would be helpful to express it in terms of the average firing rate. If you do that, lots of terms go away.

10. Figure caption 2c: "colored lines show the estimated minimal decoding time from simulations and the black lines show the fitted theoretical predictions". Presumably the description of the fitted lines was given on on l. 178. However, we can't tell from this description what you actually did. Equations are needed.

And presumably this also comes up on l. 211: "ratios of fitted regressors K1 were approximately 1.69 and 1.72 for D = 1 and D = 2, respectively".

11. l. 179: "Within this range of scale factors". Within what range?

12. l 184-5: "As suggested by Figure 2d, there is also a strong correlation between Fisher information and minimal decoding time again indicating a speed-accuracy trade-off." There's certainly a tradeoff between Fisher info and minimal decoding time, but, as mentioned in point 1, minimal decoding time isn't always a great measure. You should be up front about this.

13. The tables should be moved to the main text -- the parameters (especially the number of modules) are highly relevant. Or what would be even more useful would be to put the parameters in the figure captions.

14. Figure 5: by "Gaussian pop." and "Periodic pop." do you mean "Single peaked" and "multi-peaked"? If so, this should be clear.

15. l 303: "Thus, minimal decoding time should set a bound on the number of features a population can jointly encode reliably." Very interesting prediction! Which you point out on lines 325-7: "Our work gives a compelling reason to understand whether and how biological brains can reliably encode high-dimensional stimuli at behaviorally relevant time scales." But this should be quantified; see point 1.

*Reviewer #2 (Recommendations for the Authors):*

The paper makes a simple but important point. It is also remarkably well explained especially considering the complexity of the calculations. For once, I do not feel the need to make many suggestions for improvement.

One point that could be better addressed regards the justification of speed of encoding based on data on sensory cortex. The introduction seems heavy on the visual system based on the results of Thorpe and colleagues. In the discussion (lines 288-289) the authors point out (though the sentence is not very strongly linked to the overall text flow) that the first few spikes carry significant information in the visual and olfactory system. While the point of raising the importance f speed of encoding in sensory areas is valid, I am not sure that it should privilege the visual system or the visual and olfactory system. I think that somatosensory modalities encode information faster and earlier than visual modalities, including with regard to the prominence of first spikes (see e.g. Panzeri et al. Neuron 2001). If anything, visual cortical data show encoding in longer time scales. It would be nice to have a more precise introduction to and discussion of these topics to motivate this work and to evaluate its implications.

---

## [Author Response]

Essential revisions:While this paper makes an extremely important point, one that should definitely be communicated to the broader community, in our opinion it suffers from two (relatively easily fixable) problems:I. Unless one is an expert, it's very hard to extract why multi-peaked tuning curves lead to catastrophic errors.II. It's difficult to figure out under what circumstances multi-peaked tuning curves are bad. This is important, because there are a lot of neurons in sensory cortex, and one would like to know whether periodic tuning curves are really a bad idea there.And here are the fixes:I. Figure 1c seems like a missed opportunity to explain what's really going on, which is that on any particular trial the positions of the peaks of the log likelihood can shift in both phase and amplitude (with phase being more important). The reason it's a missed opportunity is that Figure 1c shows the average log likelihood, which makes it hard to understands why there are catastrophic errors. It would really help if Figure 1c were expanded into its own large figure, with sample log likelihoods showing catastrophic errors for multi-peaked tuning curves but not for single-peaked ones. You could also indicate why, when multi-peaked tuning curves do give the right answer, the error tends to be small.

We thank the reviewer for this suggestion. We have now split the first figure into two.

In the new Figure 1, we provide an intuitive explanation of local vs catastrophic errors and single-peaked / periodic tuning curves. We have also added smaller panels to illustrate how the distribution of errors changes with decoding time (using a simulated single-peaked population).

The new Figure 2 shows sampled likelihoods for 3 different populations. We hope this provides some intuitive understanding of the phase shifts. Unfortunately, it proved difficult not to normalize the “height” of each module’s likelihood as they can differ by several orders of magnitude across the modules. However, due to the multiplication, the peak likelihood values can (approximately) be disregarded in the ML-decoding. Lastly, we have also added more simulation points (scale factors) compared to what we had in the earlier version of the figure (see panels d-e).

II. What the reader really wants to know is: would sensory processing in real brains be more efficient if multi-peaked tuning curves were used? That's certainly hard to answer in all generality, but you could make a comparison between a code with single-peaked tuning curves and a _good_ code with multi-peaked tuning curves. Our guess is that a good code would have λ_1=1 and c around 0.5 (but not, of course, exactly 1/2; you could use the module ratio the grid cell people came up with -- we think 1/sqrt(2) -- although we doubt if it matters much). What would be great would be a relatively simple relation, possibly fitted empirically to simulations, that told us when multi-peaked tuning curves are better than single-peaked one, given the number of neurons, background and peak firing rates, and dimensionality. Of course, this may be impossible, but if so it's important to state that up front, so that people know how difficult this is.

We thank the reviewer for this comment and the suggestions. We agree, ideally such an expression would be useful. However, as you note it is a very challenging task due to the large parameter space (number of neurons, peak amplitude, ongoing background firing rate, width of tuning, stimulus dimensionality etc) and is beyond the scope of the present study. We have instead included a new figure (see Figure 7 in the manuscript) detailing the minimal decoding times for various choices of parameter values. We believe this gives an indication to how minimal decoding time scales with various parameters.

In addition, we have a number of comments, all designed to improve the paper. Feel free to disagree with any of them; we don't feel massively strongly about them. Also, if you implement them, no reason to provide a reply -- we're sensitive to the fact that replies to the reviewer are becoming longer than the paper!Reviewer #1 (Recommendations for the authors):1. When catastrophic errors are possible, we agree that it's very hard to find a good measure of error. RMS error clearly isn't very good. However, it's not clear minimal decoding time is any better, since it'd defined in terms of RMS error. Finding a good measure seems critical when comparing codes, given that Fisher info can change by 5 orders of magnitude while decoding time changes by only 1 (e.g., Figure 4d). Moreover, we could be wrong, but our intuition is that when λ_1=1, catastrophic errors aren't much of an issue. Is that correct? If so, time to threshold (defined to be time for the RMS error to be twice that predicted by the inverse of the Fisher info) could be a bad measure.Here we don't have any really great suggestions. But you might try something like [<(error)^k>/((2k-1)!!)]^{1/k} for k larger than 2 and even (the factor of (2k-1)!!) makes this equal to the RMS error for Gaussian distributions. But other measures are possible.

We thank the reviewer for these questions, they greatly helped us devise some additional figures for explanation. In the following we provide a reply to each of the points raised by the reviewer.

i. All populations (even single-peaked) suffer from catastrophic errors when the decoding time is too short.We have added a panel showing this for single-peaked tuning, see Figure 1c-e.

ii. We acknowledge that the choice of measure for minimal decoding time is not a perfect measure, but wedo not agree that it is not better than only studying RMSE. We believe they all have their strengths and weaknesses. The choice of minimal decoding time criteria compares the empirical RMSE with the predicted RMSE if no catastrophic distortion is present (i.e., inverse of Fisher information). Thus, this measure does provide new information that neither RMSE nor Fisher information in themselves capture. We have highlighted this in points iv-v.

iii. Regarding the order of magnitude, we believe that although the minimal decoding time differs only onone order of magnitude, ecologically it could be more beneficial to reduce the decoding time with (say) 30 ms per neural population while still providing reasonable accuracy.

iv. Regarding catastrophic errors for λ_1 = 1, we have added a comparison between single-peakedtuning curves and periodic tuning curves (λ_1 = 1, and c = 1/1.44) using the optimal scale factor for a 2-dimensional stimulus (Figure 7 a-c). It clearly shows how large estimation errors occur in the periodic population even after the periodic population has lower MSE than the single-peaked. However, only the single-peaked population have approached the CR-bound. This provides an example of when RMSE and Fisher information alone can provide a possibly erroneous conclusion.

v. About minimal decoding time being the wrong measure, we have also added a supplementary figureusing an alternative definition of minimal decoding time – instead of comparing MSE with the CR-bound, we compared the entire distribution of empirical errors against the Gaussian distribution predicted by Fisher information. Using one-sided Kolmogorov-Smirnov (KS) tests, we gradually increased the decoding time until the empirical distribution could not be distinguished from the predicted distribution with a significance level of 0.05. To correct for multiple sequential hypothesis testing, we imposed a Bonferroni-type of correction, scaling the significance level with the number of so-far considered decoding times. I.e., 0.05 / j, where “j” is the j:th comparison. This correction does not seem to change the results significantly and it is comparable to the value without the correction. This alternative definition gives qualitatively similar results (see Figure 3 – —figure supplement 4). [See lines 269-274]

Thus, we believe that all measures, MSE, Fisher information and minimal decoding time, have their strengths and weaknesses – therefore we argue they should be considered together and not in isolation!

2. Are you using periodic boundary conditions? Tuning curves are clearly periodic, but decoding near the boundaries is a bit tricky. For instance, the ML estimator can put the decoded value outside the range [0, 1). This should be clarified.

Yes, thank you for catching this. We assume that the stimulus is periodic in the range [0,1) for precisely this reason. We have updated the manuscript to clarify this [Line 96-99].

3. It should be clear that the expression for Fisher info (Equation 3) applies when a_i=a and b=0.

Thank you for noticing this. We have specified it in the text [Line 110]

4. Figure 1a: it would be helpful to indicate the location of s_1 and s_2 in the top panel.

Thank you for noticing this. We have mentioned it in the figure.

Also, isn't Figure 1a a bit misleading? The problem isn't a small number of spikes; it's a phase shift in the log likelihood.

The catastrophic errors are not just a property of Bayesian estimators, such errors are observed for any non-local estimator. This type of illustration as shown in our Figure is well established as an intuitive explanation of these errors (see Shannon, Sreevivasan, Wernersson, etc.). When presenting this work, we have found that this illustration is very useful for providing some intuition about the role of catastrophic errors. Secondly, regarding the number of spikes and phase shift, our study suggests that the phase shift is related to the number of spikes (see Figure 3 – —figure supplement 5). If the number of spikes is small, then the ambiguousness of a multi-peaked code might not be resolved. Thus, these two concepts are highly linked.

5. l 99-100: "For the largest mode of the joint likelihood function to also be centered close to the true stimulus condition, the distance δ between s^(1)_ML and s^(2)_ML must be smaller than between any other pair of modes of Q_1 and Q_2." We can see why this might be reasonable, but it's not totally obvious that the distance between the modes is the only thing that matters. This needs to be shown.

We thank the reviewer for the suggestion. We have clarified this in the revised manuscript. See the new text in the appendix with a reference to it in the method section, but it is (of course) an approximation. [line 574-577 and Appendix 3]

6. l 100-101: "Thus, to avoid catastrophic errors, δ must be smaller than some largest allowed distance δ* (see Methods for calculation of δ*)." This makes it seems like δ* is something complicated. But in fact it's exactly half the smallest distance between any other pair of modes. Why not say so?

Strictly speaking δ* is not half the distance of any other pair of peaks.\δ* depends on \δ_{0,0} (see equations 11-16). It is unfortunately not easy to describe it in words.

7. l 120-1: "estimates can cause catastrophic shifts in the joint likelihood function. For single-peaked tuning curves (λ_1 = 1), however, only small scale factors c can pose such problems." Should "For single-peaked tuning curves" be "When the first mode has a single peak". If not, we're lost.

Yes, we agree with the reviewer. We have, corrected this [see Line 183]

8. l 128: "Furthermore, only populations with unambiguous codes over the stimulus interval were included [22]." Can't parse.

We have added a clarification that the only requirement is that c is not ½, ⅓, ¼ etc. [see Lines 196-197]

9. Equation 8 seems unlikely to be correct, given that it doesn't depend on c, which we know is critical (as shown in Figure 1e). However, if you do keep it, it would be helpful to express it in terms of the average firing rate. If you do that, lots of terms go away.

Equation 8 is correct, we clarified that it is sample averages and that the scale factor is implicitly defined within this average. We also added the suggested change, but as this is (strictly speaking) only true for integer number of peaks – we opted to add it as an approximate equation. If not integer peaks, the average firing rate is not exactly this expression for homogeneous amplitudes a_i = a. As it turns out in simulations, though, it is approximately true. [See lines 222-225]

10. Figure caption 2c: "colored lines show the estimated minimal decoding time from simulations and the black lines show the fitted theoretical predictions". Presumably the description of the fitted lines was given on on l. 178. However, we can't tell from this description what you actually did. Equations are needed.And presumably this also comes up on l. 211: "ratios of fitted regressors K1 were approximately 1.69 and 1.72 for D = 1 and D = 2, respectively".

We have added equation references for this. [Line 259]

11. l. 179: "Within this range of scale factors". Within what range?

We have changed the text to “Within the simulated range of scale factors” [Line 259]

12. l 184-5: "As suggested by Figure 2d, there is also a strong correlation between Fisher information and minimal decoding time again indicating a speed-accuracy trade-off." There's certainly a tradeoff between Fisher info and minimal decoding time, but, as mentioned in point 1, minimal decoding time isn't always a great measure. You should be up front about this.

See previous comments.

13. The tables should be moved to the main text -- the parameters (especially the number of modules) are highly relevant. Or what would be even more useful would be to put the parameters in the figure captions.

Thank you for this suggestion, we have done so for Figures 1-4. For the spiking model, we have not included any equations in the result section and therefore we are not sure adding the parameters would make sense. Therefore, we kept the tables in the method section for Figures 5-6.

14. Figure 5: by "Gaussian pop." and "Periodic pop." do you mean "Single peaked" and "multi-peaked"? If so, this should be clear.

Thank you for catching this oversight. We have corrected this. See Figures 5-6

15. l 303: "Thus, minimal decoding time should set a bound on the number of features a population can jointly encode reliably." Very interesting prediction! Which you point out on lines 325-7: "Our work gives a compelling reason to understand whether and how biological brains can reliably encode high-dimensional stimuli at behaviorally relevant time scales." But this should be quantified; see point 1.

We thank the review for this comment. We hope the new Figure 7 will clarify this.

Reviewer #2 (Recommendations for the Authors):The paper makes a simple but important point. It is also remarkably well explained especially considering the complexity of the calculations. For once, I do not feel the need to make many suggestions for improvement.One point that could be better addressed regards the justification of speed of encoding based on data on sensory cortex. The introduction seems heavy on the visual system based on the results of Thorpe and colleagues. In the discussion (lines 288-289) the authors point out (though the sentence is not very strongly linked to the overall text flow) that the first few spikes carry significant information in the visual and olfactory system. While the point of raising the importance f speed of encoding in sensory areas is valid, I am not sure that it should privilege the visual system or the visual and olfactory system. I think that somatosensory modalities encode information faster and earlier than visual modalities, including with regard to the prominence of first spikes (see e.g. Panzeri et al. Neuron 2001). If anything, visual cortical data show encoding in longer time scales. It would be nice to have a more precise introduction to and discussion of these topics to motivate this work and to evaluate its implications.

Thank you, this is an important point. We agree that sensory modalities ought to have similar constraints, and might operate on even faster time scales. Our results might also be applicable outside of the visual areas, too. We focused on early visual processing due to the established smooth single-peaked tuning and the (often) periodic sensory features (e.g., orientation and color). For other modalities, we are not sure the tuning is continuous (e.g., odor) or periodic (e.g., somatosensory stimuli). All the simulations assumed periodic stimuli, however, we do not believe the periodic stimulus is of great importance for our results. For discrete stimuli, catastrophic errors can also occur – but the calculations made here assumed continuous variables. Extending this work to discrete stimulus points is an interesting direction.

We have expanded more on the applicability outside of the visual area in the discussion. [Lines 411, 446-450].